# Impact of vaccination on the association of COVID-19 with cardiovascular diseases: An OpenSAFELY cohort study

Genevieve I. Cezard [1,2,25], Rachel E. Denholm[3,4,5,25], Rochelle Knight [3,4,6,7,25], Yinghui Wei [8], Lucy Teece[9], Renin Toms [3,10], Harriet J. Forbes[11], Alex J. Walker [12], Louis Fisher[12], Jon Massey [12], Lisa E. M. Hopcroft [13], Elsie M. F. Horne[3,4], Kurt Taylor[3], Tom Palmer [3,6], Marwa Al Arab[3], Jose Ignacio Cuitun Coronado[3], Samantha H. Y. Ip [1,2,14], Simon Davy[12], Iain Dillingham[12], Sebastian Bacon[12], Amir Mehrkar [12], Caroline E. Morton[15], Felix Greaves[16,17], Catherine Hyams [3], George Davey Smith [3,6], John Macleod[3,5,7], Nishi Chaturvedi [18], Ben Goldacre[12], William N. Whiteley [19], Angela M. Wood [1,2,20,21,22,23,25], Jonathan A. C. Sterne [3,4,5,25] ✉ & Venexia Walker [3,6,24,25]  On behalf of the Longitudinal Health and Wellbeing and Data and Connectivity UK COVID-19 National Core Studies, CONVALESCENCE study and the OpenSAFELY collaborative

Infection with SARS-CoV-2 is associated with an increased risk of arterial and venous thrombotic events, but the implications of vaccination for this increased risk are uncertain. With the approval of NHS England, we quantified associations between COVID-19 diagnosis and cardiovascular diseases in different vaccination and variant eras using linked electronic health records for ~40% of the English population. We defined a 'pre-vaccination' cohort (18,210,937 people) in the wild-type/Alpha variant eras (January 2020-June 2021), and 'vaccinated' and 'unvaccinated' cohorts (13,572,399 and 3,161,485 people respectively) in the Delta variant era (June-December 2021). We showed that the incidence of each arterial thrombotic, venous thrombotic and other cardiovascular outcomes was substantially elevated during weeks 1-4 after COVID-19, compared with before or without COVID-19, but less markedly elevated in time periods beyond week 4. Hazard ratios were higher after hospitalised than non-hospitalised COVID-19 and higher in the pre-vaccination and unvaccinated cohorts than the vaccinated cohort. COVID-19 vaccination reduces the risk of cardiovascular events after COVID-19 infection. People who had COVID-19 before or without being vaccinated are at higher risk of cardiovascular events for at least two years.

Infection with severe acute respiratory syndrome coronavirus 2 (SARS-CoV-2) increases the risk of arterial thrombotic events (ATE), such as myocardial infarction (MI) and ischaemic stroke, and venous thrombotic events (VTE), such as pulmonary embolism (PE) and lower limb deep vein thrombosis (DVT)[1]. This elevation in risk is highest immediately after infection, and higher after severe COVID-19[1].

The dominant SARS-CoV-2 variant changed during the pandemic, and different variants may be associated with different subsequent

risks of major vascular events. The Delta variant (B.1.617.2), which emerged in late 2020 and was dominant by mid-2021[2–6], was associated with a greater risk of hospitalisation and death than the Alpha variant[7–11].

The UK's COVID-19 vaccine rollout started on December 8th, 2020, with eligibility in order of priority groups determined by the Joint Committee on Vaccination and Immunisation (JCVI), based on age, clinical vulnerability and health and social care occupation[12]. All adults in England became eligible to receive a first vaccination by June 18th 2021, and a second vaccination by August 2021[13].

Cohort studies have reported lower incidence of arterial and venous thrombotic events after COVID-19 vaccination than in unvaccinated people[14] and that rates of hospitalisation for acute MI and ischemic stroke 31-120 days after COVID-19 are lower in vaccinated than unvaccinated people[15]. Randomised trials suggest that seasonal influenza vaccination reduces rates of cardiovascular events and cardiovascular death[16]. However, uncertainties remain. In younger people, the rare complications of the Pfizer–BioNTech BNT162b2 and Moderna mRNA-1273 mRNA vaccines (myocarditis)[17,18] and the Oxford–AstraZeneca ChAdOx1 nCoV-19 AZD1222 vaccine (vaccine induced thrombotic thrombocytopenia)[19] are balanced by reduced risk of severe COVID-19, but the role of these vaccines in modifying the incidence of common vascular events after COVID-19 is less clear.

Using linked anonymised electronic health records on 18.2 million adults registered with English general practices (GP), we compared the incidence of vascular diseases after COVID-19 with the incidence before or without COVID-19, in a pre-vaccination cohort followed during the wild-type and Alpha variant eras; and in vaccinated and unvaccinated cohorts followed during the Delta variant era (Fig. 1). Differences between associations in the pre-vaccination and unvaccinated cohorts should relate to effects of changing variants, while differences between the vaccinated and unvaccinated cohorts should relate to the effect of vaccination.

## Results
### Characteristics of study cohorts
Among 18,210,937 people in the pre-vaccination cohort, 1,150,299 had a COVID-19 diagnosis during follow-up of whom 75,667 (6.6%) were hospitalised. There were 844,235 COVID-19 diagnoses (15,342 (1.8%) hospitalised) among 13,572,399 people in the vaccinated cohort and 162,103 (9,250 (5.7%) hospitalised) among 3,161,485 people in the unvaccinated cohort (Table 1). Among 75,667 people in the pre-vaccination cohort who were hospitalised because of COVID-19 37,881 (50.1%) were hospitalised on the day of diagnosis and 6,278 (8.3%) were hospitalised the day after diagnosis: the remainder were hospitalised 2–28 days after diagnosis (Table S1). In the pre-vaccination cohort, the median age was 49 years (interquartile range (IQR) 34–64), a slight majority (50.2%) were female, and 78.0%, 6.4% and 2.2% were recorded as being White, South Asian and Black ethnicities respectively. Differences between the vaccinated and unvaccinated cohorts reflected predictors of COVID-19 vaccine uptake[20]. The median (IQR) age was 54 (IQR 39–68) years in the vaccinated cohort, compared with 36 (IQR 28–47) years in the unvaccinated cohort. The proportions of females were 52.1% and 42.0% in the vaccinated and unvaccinated cohorts respectively, while the proportions recorded as of White ethnicity were 82.1% and 61.6% respectively, and the proportions living in the most

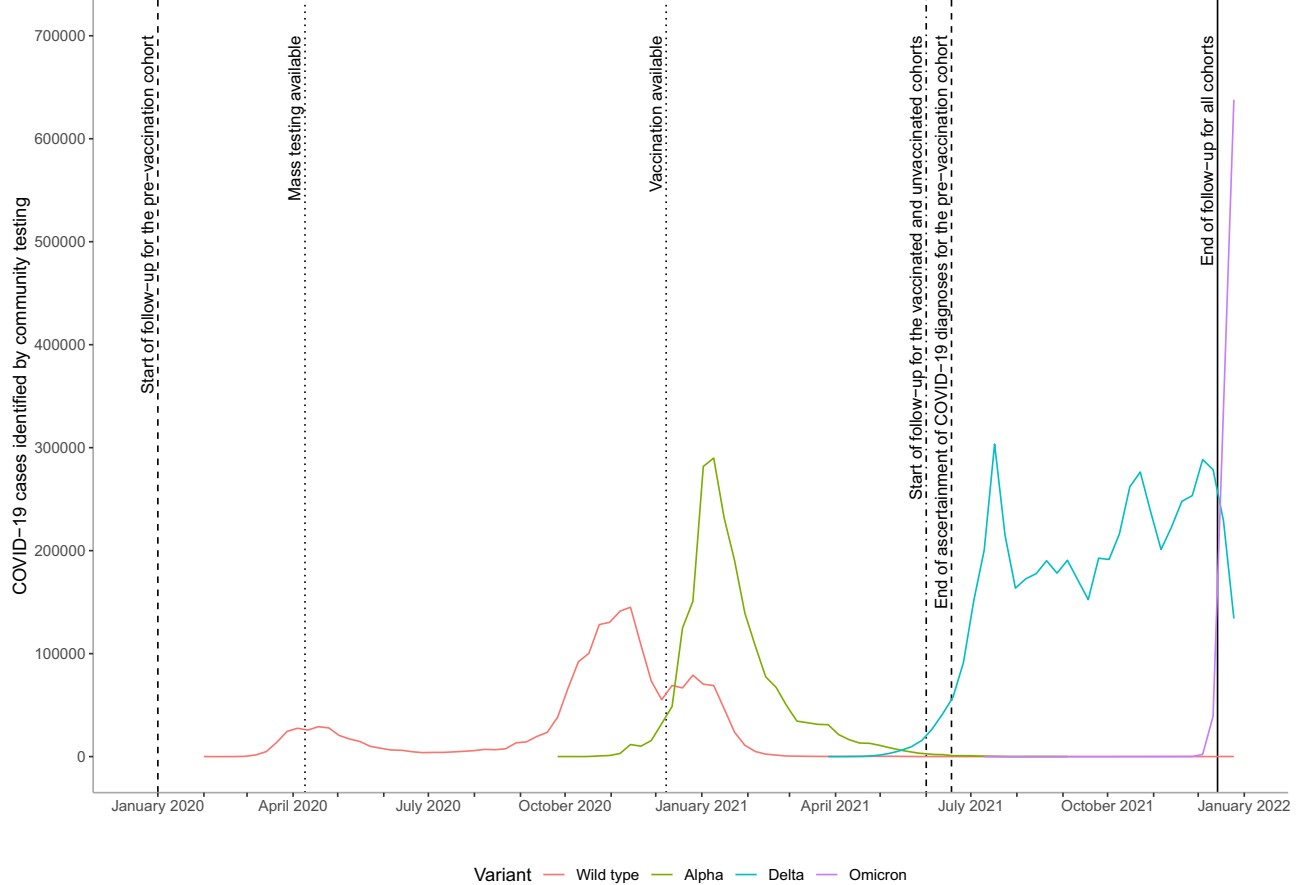

**Fig. 1 | Estimated number of COVID-19 cases identified by community testing in England between January 2020 and January 2022.** The start and end of follow-up for each cohort, the periods during which different variants were dominant and the times that mass testing became available and vaccination rollout commenced are shown. Code and underlying data are available in the following repository: https://github.com/grpEHR/covid_variants.

**Table 1 | Patient characteristics in the pre-vaccination, vaccinated and unvaccinated cohorts**

| Characteristics | | Pre-vaccination cohort (Jan 1 2020 to Dec 14 2021) | | Vaccinated cohort (June 1 to Dec 14 2021) | | Unvaccinated cohort (June 1 to Dec 14 2021) | |
|---|---|---|---|---|---|---|---|
| | | N (%) | COVID-19 diagnoses | N (%) | COVID-19 diagnoses | N (%) | COVID-19 diagnoses |
| All | | 18,210,937 | 1,150,299 | 13,572,399 | 844,235 | 3,161,485 | 162,103 |
| Sex | Female | 9,144,881 (50.2%) | 627,664 | 7,069,546 (52.1%) | 469,430 | 1,328,645 (42.0%) | 86,431 |
| | Male | 9,066,056 (49.8%) | 522,635 | 6,502,853 (47.9%) | 374,805 | 1,832,840 (58.0%) | 75,672 |
| Age, years | 18–29 | 3,202,070 (17.6%) | 261,713 | 1,676,216 (12.4%) | 92,584 | 943,544 (29.8%) | 43,264 |
| | 30–39 | 3,161,886 (17.4%) | 220,784 | 1,885,074 (13.9%) | 145,292 | 945,077 (29.9%) | 54,772 |
| | 40–49 | 2,988,406 (16.4%) | 207,475 | 2,136,224 (15.7%) | 229,337 | 601,890 (19.0%) | 36,422 |
| | 50–59 | 3,183,224 (17.5%) | 203,992 | 2,632,224 (19.4%) | 198,393 | 368,485 (11.7%) | 18,297 |
| | 60–69 | 2,485,644 (13.6%) | 105,671 | 2,238,772 (16.5%) | 102,758 | 184,888 (5.8%) | 6092 |
| | 70–79 | 2,011,543 (11.0%) | 68,179 | 1,927,005 (14.2%) | 53,150 | 79,879 (2.5%) | 2045 |
| | 80–89 | 968,302 (5.3%) | 59,078 | 891,437 (6.6%) | 18,148 | 29,663 (0.9%) | 968 |
| | 90+ | 209,862 (1.2%) | 23,407 | 185,447 (1.4%) | 4573 | 8,059 (0.3%) | 243 |
| Ethnicity | White | 14,199,514 (78.0%) | 871,142 | 11,146,222 (82.1%) | 716,017 | 1,947,330 (61.6%) | 122,402 |
| | Mixed | 211,010 (1.2%) | 15,545 | 122,352 (0.9%) | 7334 | 77,925 (2.5%) | 4074 |
| | South Asian | 1,168,451 (6.4%) | 129,517 | 762,495 (5.6%) | 40,741 | 314,718 (10.0%) | 10,718 |
| | Black | 394,135 (2.2%) | 30,176 | 210,470 (1.6%) | 9379 | 168,355 (5.3%) | 8227 |
| | Other | 408,705 (2.2%) | 22,200 | 224,004 (1.7%) | 10,288 | 181,162 (5.7%) | 4351 |
| | Missing | 1,829,122 (10.0%) | 81,719 | 1,106,856 (8.2%) | 60,476 | 471,995 (14.9%) | 12,331 |
| Index of multiple deprivation quintile | 1: Most deprived | 3,525,620 (19.4%) | 279,926 | 2,222,818 (16.4%) | 133,651 | 941,492 (29.8%) | 49,101 |
| | 2 | 3,630,499 (19.9%) | 249,597 | 2,549,290 (18.8%) | 156,080 | 760,767 (24.1%) | 38,474 |
| | 3 | 3,938,026 (21.6%) | 231,601 | 3,000,854 (22.1%) | 181,721 | 634,857 (20.1%) | 32,075 |
| | 4 | 3,710,471 (20.4%) | 208,451 | 2,959,981 (21.8%) | 186,328 | 484,726 (15.3%) | 24,590 |
| | 5: Least deprived | 3,406,321 (18.7%) | 180,724 | 2,839,456 (20.9%) | 186,455 | 339,643 (10.7%) | 17,863 |
| Smoking status | Never smoker | 8,362,691 (45.9%) | 566,479 | 6,313,346 (46.5%) | 411,248 | 1,312,958 (41.5%) | 65,184 |
| | Former smoker | 5,973,167 (32.8%) | 385,150 | 4,947,123 (36.4%) | 324,969 | 632,484 (20.0%) | 47,422 |
| | Current smoker | 3,119,113 (17.1%) | 154,428 | 1,904,804 (14.0%) | 91,978 | 858,703 (27.2%) | 40,502 |
| | Missing | 755,966 (4.2%) | 44,242 | 407,126 (3.0%) | 16,040 | 357,340 (11.3%) | 8995 |
| Region | East | 4,222,601 (23.2%) | 260,099 | 3,188,143 (23.5%) | 179,633 | 710,839 (22.5%) | 36,946 |
| | East Midlands | 3,163,448 (17.4%) | 215,843 | 2,378,711 (17.5%) | 155,625 | 514,154 (16.3%) | 30,669 |
| | London | 1,250,376 (6.9%) | 77,260 | 718,413 (5.3%) | 37,512 | 458,007 (14.5%) | 12,324 |
| | North East | 882,703 (4.8%) | 68,550 | 656,311 (4.8%) | 48,974 | 135,739 (4.3%) | 7890 |
| | North West | 1,613,534 (8.9%) | 122,524 | 1,230,083 (9.1%) | 85,666 | 213,099 (6.7%) | 12,999 |
| | South East | 1,238,689 (6.8%) | 60,689 | 952,166 (7.0%) | 56,305 | 195,133 (6.2%) | 9821 |
| | South West | 2,543,525 (14.0%) | 94,757 | 2,095,323 (15.4%) | 126,893 | 313,208 (9.9%) | 19,032 |
| | West Midlands | 742,813 (4.1%) | 62,997 | 484,928 (3.6%) | 29,502 | 174,477 (5.5%) | 8681 |
| | Yorkshire/Humber | 2,553,248 (14.0%) | 187,580 | 1,868,321 (13.8%) | 124,125 | 446,829 (14.1%) | 23,741 |
| Care home resident | | 89,607 (0.5%) | 23,916 | 58,256 (0.4%) | 2883 | 2988 (0.1%) | 121 |

deprived areas were 16.4% and 29.8% respectively. Compared with the vaccinated cohort, people in the unvaccinated cohort were more likely to be smokers, less likely to consult their GPs and less likely to have prior medical problems recorded (Table S2).

**Number of events and incidence rates**
The numbers of events, person-years, and incidence rates per 100,000 person-years of vascular events before any COVID-19 diagnosis, after hospitalised COVID-19 and after non-hospitalised COVID-19, are presented, for each cohort and outcome, in Table 2. There was a total of 212,557, 57,425 and 3,316 ATE in the pre-vaccination, vaccinated and unvaccinated cohorts respectively. The corresponding total numbers of VTE were 117,730, 29,107 and 3,178 respectively. In each cohort, the incidence of each arterial thrombotic and venous thrombotic event was higher after COVID-19 than before or without COVID-19. For each

outcome and cohort, the highest incidence rates were after hospitalised COVID-19. Incidence rates were generally lower in the unvaccinated cohort than in the vaccinated cohort, as expected given that the median age of the unvaccinated cohort (36 years) was much lower than that of the vaccinated cohort (54 years).

**Comparisons of event rates after COVID-19 diagnosis versus before or without COVID-19 diagnosis**
Adjusted hazard ratios were estimated using Cox-proportional hazards models to quantify the associations between COVID-19 diagnosis (time-varying exposure) and a first cardiovascular event including arterial thrombotic, venous thrombotic and other cardiovascular events. Minimally adjusted models accounted for age, sex and region and maximally adjusted models accounted additionally for ethnicity, area deprivation, smoking status, number of GP-patient interactions

**Table 2 | Number of cardiovascular events in the pre-vaccination cohort, vaccinated cohort and unvaccinated cohort, with person-years of follow-up, by COVID-19 severity**

| Event | COVID-19 severity | Pre-vaccination cohort (N = 18,210,937) | | Vaccinated cohort (N = 13,572,399) | | Unvaccinated cohort (N = 3,161,485) | |
|---|---|---|---|---|---|---|---|
| | | Event/person-years | Incidence rate* | Event/person-years | Incidence rate* | Event/person-years | Incidence rate* |
| **Arterial thrombotic events** | | | | | | | |
| All arterial thrombotic events | No COVID-19 | 203,473/33,995,522 | 599 | 55,521/6,192,325 | 897 | 2951/1,112,947 | 265 |
| | Hospitalised COVID-19 | 2304/47,431 | 4858 | 407/2,679 | 15,191 | 169/1649 | 10,250 |
| | Non-hospitalised COVID-19 | 6780/966,400 | 702 | 1497/155,606 | 962 | 196/23,926 | 819 |
| Acute myocardial infarction | No COVID-19 | 97,236/34,076,517 | 285 | 25,360/6,199,559 | 409 | 1432/1,113,291 | 129 |
| | Hospitalised COVID-19 | 1150/48,545 | 2369 | 197/2717 | 7252 | 72/1,661 | 4335 |
| | Non-hospitalised COVID-19 | 3116/970,499 | 321 | 677/155,813 | 434 | 86/23,952 | 359 |
| Ischaemic stroke | No COVID-19 | 100,350/34,072,265 | 295 | 28,194/6,198,458 | 455 | 1377/1,113,278 | 124 |
| | Hospitalised COVID-19 | 1052/48,415 | 2173 | 176/2716 | 6481 | 74/1662 | 4453 |
| | Non-hospitalised COVID-19 | 3517/970,028 | 363 | 799/155,793 | 513 | 93/23,949 | 388 |
| **Venous thrombotic events** | | | | | | | |
| All venous thrombotic events | No COVID-19 | 108,798/34,065,305 | 319 | 27,303/6,198,810 | 440 | 2265/1,113,091 | 203 |
| | Hospitalised COVID-19 | 3868/45,938 | 8420 | 781/2615 | 29,870 | 658/151 | 42,427 |
| | Non-hospitalised COVID-19 | 5064/968,526 | 523 | 1023/155,742 | 657 | 255/23,918 | 1066 |
| Pulmonary embolism | No COVID-19 | 46,592/34,116,905 | 137 | 11,322/6,202,790 | 183 | 755/1,113,435 | 68 |
| | Hospitalised COVID-19 | 3093/46,783 | 6611 | 663/2639 | 25,120 | 587/1566 | 37,481 |
| | Non-hospitalised COVID-19 | 2616/971,311 | 269 | 533/155,882 | 342 | 170/23,943 | 710 |
| Deep vein thrombosis | No COVID-19 | 58,125/34,099,045 | 170 | 14,308/6,201,621 | 231 | 1376/1,113,277 | 124 |
| | Hospitalised COVID-19 | 826/48,540 | 1702 | 124/2723 | 4554 | 76/1657 | 4588 |
| | Non-hospitalised COVID-19 | 2319/971,257 | 239 | 450/155,863 | 289 | 85/23,949 | 355 |
| **Other cardiovascular events** | | | | | | | |
| Heart failure | No COVID-19 | 322,074/33,893,733 | 950 | 126,530/6,173,040 | 2050 | 4,280/1,112,569 | 385 |
| | Hospitalised COVID-19 | 4998/44,622 | 11,201 | 1,591/2,399 | 66,313 | 303/1621 | 18,693 |
| | Non-hospitalised COVID-19 | 8595/963,587 | 892 | 2375/155,244 | 1530 | 152/23,933 | 635 |
| Angina | No COVID-19 | 238,660/33,922,822 | 704 | 81,037/6,182,983 | 1311 | 2530/1,112,942 | 227 |
| | Hospitalised COVID-19 | 2686/46,168 | 5818 | 864/2545 | 33,952 | 166/1634 | 10,160 |
| | Non-hospitalised COVID-19 | 5675/965,281 | 588 | 1452/155,417 | 934 | 88/23,943 | 368 |
| Transient ischaemic attack | No COVID-19 | 46,617/34,105,801 | 137 | 12,707/6,201,839 | 205 | 463/1,113,466 | 42 |
| | Hospitalised COVID-19 | 246/49,114 | 501 | 45/2736 | 1645 | 7/1669 | 419 |
| | Non-hospitalised COVID-19 | 1070/972505 | 110 | 222/155,906 | 142 | 13/23,967 | 54 |
| Subarachnoid haemorrhage / haemorrhagic stroke | No COVID-19 | 18,308/34,137,474 | 54 | 4389/6,204,415 | 71 | 295/1,113,538 | 26 |
| | Hospitalised COVID-19 | 222/49,235 | 451 | 24/2,743 | 875 | 9/1670 | 539 |
| | Non-hospitalised COVID-19 | 782/973,084 | 80 | 169/155,958 | 108 | 16/23,967 | 67 |

*Incidence rates are per 100,000 person-years.

and history of comorbidities. In each cohort, maximally adjusted HRs (aHRs) comparing the incidence of each outcome after COVID-19 diagnosis with the incidence before or without COVID-19 diagnosis were attenuated compared with age-, sex- and region-adjusted HRs (Tables 3-5, Table S3). For all outcomes, hazard ratios were extremely high on the day of COVID-19 diagnosis (day 0), particularly among individuals hospitalised with COVID-19 on the day of diagnosis. The incidence of each outcome in each cohort was also elevated during weeks 1-4 after COVID-19 diagnosis, compared with before or without COVID-19 diagnosis. aHRs were lower in subsequent time periods than during weeks 1-4 after COVID-19 diagnosis, though they were generally greater than 1 throughout follow-up in each cohort (Figs. 2 and 3, Tables 3-5). aHRs during weeks 1-4 after COVID-19 diagnosis were substantially lower in the vaccinated cohort than in the pre-vaccination and unvaccinated cohorts, and generally remained lower than in other cohorts during weeks 4-28 (Figs. 2 and 3, Tables 3-5). For each outcome and in each cohort, aHRs were substantially higher after hospitalised than non-hospitalised COVID-19 (Fig. 2, Tables 3 and 4, Table S3).

The incidence of ATE during weeks 1-4 after COVID-19 diagnosis, compared with before or without COVID-19 diagnosis, was elevated in the pre-vaccination and unvaccinated cohorts (aHRs 4.40 (95% CI 4.16–4.65) and 8.53 (7.20–10.1) respectively) but less markedly elevated in the vaccinated cohort (2.09 (1.92–2.28)) (Fig. 2, Table 3). The

incidence of ATE remained elevated during weeks 5-28 in the unvaccinated cohort (1.54 (1.16–2.04)) and up to weeks 53–102 in the pre-vaccination cohort (1.22 (1.14–1.30)). During weeks 1–4 the aHRs for ATE were substantially lower in the vaccinated cohort than in the unvaccinated or pre-vaccination cohorts (ratios of aHRs 0.28 (0.25–0.32) and 0.36 (0.33–0.38) respectively, Table S4). Although attenuated, aHRs remained lower in the vaccinated cohort than in the unvaccinated or pre-vaccination cohorts during weeks 5–28 (ratios of aHRs 0.70 (0.52–0.94) and 0.73 (0.66–0.82), respectively).

The aHRs for ATE were substantially higher during weeks 1–4 after hospitalised COVID-19, versus before or without COVID-19 diagnosis (pre-vaccination cohort 12.1 (11.2–13.1), unvaccinated cohort 19.6 (15.6–24.5)) than after non-hospitalised COVID-19 (pre-vaccination cohort 2.70 (2.50–2.92), unvaccinated cohort 4.35 (3.36–5.64)). In sensitivity analyses restricted to primary diagnoses of ATE, aHRs during weeks 1–4 after hospitalised COVID-19 (including both day 0 and the rest of that period) were attenuated compared with aHRs for all ATEs (Figure S1). Estimated hazard ratios were similar in sensitivity analyses removing censoring at first vaccination in the unvaccinated cohort (Table S5). In additional analyses splitting follow-up during weeks 1-4 into shorter time intervals, hazard ratios for ATE declined steadily from days 1–6 to days 21–27 after COVID-19 diagnosis, in all cohorts (Table S6).

**Table 3 | Adjusted hazard ratios (95% CI) comparing the incidence of arterial thrombotic events after versus before or without a COVID-19 diagnosis, in the pre-vaccination, vaccinated and unvaccinated cohorts, overall and according to COVID-19 severity**

| | | Time since COVID-19 diagnosis | Pre-vaccination cohort | Vaccinated cohort | Unvaccinated cohort |
|---|---|---|---|---|---|
| All arterial thrombotic events | All, age/sex/region adjusted | Day 0 | 239.9 (230.6–249.6) | 72.8 (67.9–78.1) | 254.8 (217.3–298.7) |
| | | 1–4 weeks | 5.14 (4.87–5.43) | 2.11 (1.94–2.30) | 10.5 (8.91–12.5) |
| | | 5–28 weeks | 1.55 (1.49–1.61) | 1.10 (1.01–1.20) | 1.90 (1.44–2.52) |
| | | 29–52 weeks | 1.28 (1.22–1.34) | – | – |
| | | 53–102 weeks | 1.52 (1.42–1.62) | – | – |
| | All | Day 0 | 200.0 (192.1–208.2) | 71.8 (66.9–77.0) | 198.5 (168.8–233.4) |
| | | 1–4 weeks | 4.40 (4.16–4.65) | 2.09 (1.92–2.28) | 8.53 (7.20–10.1) |
| | | 5–28 weeks | 1.35 (1.30–1.41) | 1.08 (1.00–1.18) | 1.54 (1.16–2.04) |
| | | 29–52 weeks | 1.12 (1.07–1.17) | – | – |
| | | 53–102 weeks | 1.22 (1.14–1.30) | – | – |
| | Hospitalised COVID-19 | Day 0 | 228.0 (210.6–246.7) | 123.0 (104.5–144.8) | 289.6 (224.4–373.8) |
| | | 1–4 weeks | 12.1 (11.2–13.1) | 7.44 (6.41-8.63) | 19.6 (15.6-24.5) |
| | | 5–28 weeks | 1.79 (1.63–1.96) | 1.75 (1.39-2.20) | 1.56 (0.88-2.77) |
| | | 29–52 weeks | 1.34 (1.20–1.50) | – | – |
| | | 53–102 weeks | 1.33 (1.16–1.53) | – | – |
| | Non-hospitalised COVID-19 | Day 0 | 193.5 (184.8–202.5) | 63.7 (58.9–68.9) | 152.4 (123.8–187.7) |
| | | 1–4 weeks | 2.70 (2.50–2.92) | 1.53 (1.38–1.69) | 4.35 (3.36–5.64) |
| | | 5–28 weeks | 1.28 (1.22–1.34) | 1.02 (0.93–1.11) | 1.45 (1.05–1.99) |
| | | 29–52 weeks | 1.08 (1.03–1.14) | – | – |
| | | 53–102 weeks | 1.19 (1.10–1.28) | – | – |
| Acute myocardial infarction | | Day 0 | 173.0 (162.6–184.1) | 64.5 (58.0–71.7) | 144.9 (111.8–187.8) |
| | | 1–4 weeks | 4.29 (3.96–4.64) | 1.97 (1.74–2.23) | 7.27 (5.65–9.35) |
| | | 5–28 weeks | 1.32 (1.25–1.40) | 1.05 (0.93–1.18) | 1.61 (1.11–2.35) |
| | | 29–52 weeks | 1.16 (1.09–1.24) | – | – |
| | | 53–102 weeks | 1.31 (1.19–1.45) | – | – |
| Ischaemic stroke | | Day 0 | 214.9 (203.3–227.1) | 79.9 (72.7-87.7) | 212.0 (168.7–266.4) |
| | | 1–4 weeks | 3.93 (3.62–4.27) | 2.12 (1.88–2.39) | 7.40 (5.70–9.59) |
| | | 5–28 weeks | 1.39 (1.32–1.48) | 1.09 (0.96–1.22) | 1.27 (0.81–1.98) |
| | | 29–52 weeks | 1.08 (1.01–1.15) | – | – |
| | | 53–102 weeks | 1.16 (1.05–1.27) | – | – |

Hazard ratios are maximally adjusted unless otherwise stated.

**Table 4 | Adjusted hazard ratios (95% CI) comparing the incidence of venous thrombotic events after versus before or without a COVID-19 diagnosis, in the pre-vaccination, vaccinated and unvaccinated cohorts, overall and according to COVID-19 severity**

| | | Time since COVID-19 diagnosis | Pre-vaccination cohort | Vaccinated cohort | Unvaccinated cohort |
|---|---|---|---|---|---|
| All venous thrombotic events | All, age/sex/region adjusted | Day 0 | 353.9 (338.8–369.7) | 76.1 (69.6–83.1) | 474.5 (418.4–538.1) |
| | | 1–4 weeks | 18.5 (17.8–19.2) | 5.10 (4.75–5.48) | 37.1 (33.5–41.1) |
| | | 5–28 weeks | 2.05 (1.96–2.14) | 1.61 (1.47–1.77) | 3.17 (2.50–4.00) |
| | | 29–52 weeks | 1.20 (1.13–1.27) | – | – |
| | | 53–102 weeks | 1.42 (1.29–1.56) | – | – |
| | All | Day 0 | 312.1 (298.6–326.3) | 72.3 (66.2–79.0) | 382.6 (336.8–434.6) |
| | | 1–4 weeks | 16.6 (15.9–17.2) | 4.87 (4.53–5.23) | 29.6 (26.7–32.9) |
| | | 5–28 weeks | 1.86 (1.78–1.95) | 1.53 (1.39-1.67) | 2.36 (1.87-2.99) |
| | | 29–52 weeks | 1.09 (1.03–1.16) | – | – |
| | | 53–102 weeks | 1.20 (1.09–1.32) | – | – |
| | Hospitalised COVID-19 | Day 0 | 981.5 (919.9–1047) | 436.1 (379.3–501.4) | 2426.8 (2064.3–2853.0) |
| | | 1–4 weeks | 88.5 (84.1–93.2) | 46.7 (42.2–51.6) | 199.8 (174.7–228.5) |
| | | 5–28 weeks | 4.39 (4.01–4.80) | 5.67 (4.63-6.94) | 7.50 (5.07-11.1) |
| | | 29–52 weeks | 1.39 (1.18–1.63) | – | – |
| | | 53–102 weeks | 1.27 (1.03-1.58) | – | – |
| | Non-hospitalised COVID-19 | Day 0 | 196.1 (184.9–208.0) | 45.5 (40.6–51.0) | 107.0 (85.1–134.5) |
| | | 1–4 weeks | 5.99 (5.61–6.40) | 2.13 (1.91–2.38) | 7.64 (6.34–9.20) |
| | | 5–28 weeks | 1.54 (1.47–1.63) | 1.28 (1.15-1.41) | 1.62 (1.21-2.16) |
| | | 29–52 weeks | 1.05 (0.99–1.12) | – | – |
| | | 53–102 weeks | 1.16 (1.05–1.29) | – | – |
| Pulmonary embolism | | Day 0 | 587.4 (557.5–618.8) | 137.3 (123.7–152.4) | 1151.7 (990.9–1338.5) |
| | | 1–4 weeks | 31.7 (30.3–33.1) | 9.10 (8.36-9.90) | 82.8 (72.7-94.3) |
| | | 5–28 weeks | 2.01 (1.88–2.15) | 1.75 (1.53–2.00) | 4.15 (3.01–5.74) |
| | | 29–52 weeks | 1.03 (0.94–1.13) | – | – |
| | | 53–102 weeks | 1.14 (0.99–1.32) | – | – |
| Deep vein thrombosis | | Day 0 | 123.2 (112.5–134.9) | 27.3 (22.5–33.0) | 75.5 (55.4–103.0) |
| | | 1–4 weeks | 6.65 (6.15–7.20) | 2.37 (2.07–2.72) | 6.49 (5.16–8.17) |
| | | 5–28 weeks | 1.77 (1.67–1.88) | 1.48 (1.30–1.67) | 1.61 (1.15–2.26) |
| | | 29–52 weeks | 1.17 (1.08–1.26) | – | – |
| | | 53-102 weeks | 1.28 (1.13–1.45) | – | – |

Hazard ratios are maximally adjusted unless otherwise stated.

The aHRs for VTE during weeks 1-4 after COVID-19 diagnosis, versus before or without COVID-19 diagnosis, were substantially higher than for ATE, particularly in the pre-vaccination and unvaccinated cohorts (aHRs 16.6 (95% CI 15.9–17.2) and 29.6 (26.7–32.9) respectively), but less markedly in the vaccinated cohort (4.87 (4.53–5.23)) (Fig. 2, Table 4). The incidence of VTE remained elevated, compared with before or without COVID-19 diagnosis, during weeks 5–28 in all cohorts and up to weeks 53–102 in the pre-vaccination cohort (1.20 (1.09–1.32)). During weeks 1–4 the aHRs for VTE were substantially lower in the vaccinated cohort than in the unvaccinated or pre-vaccination cohorts (ratios of aHRs 0.17 (0.15–0.19) and 0.24 (0.23–0.26) respectively, Table S4). Although attenuated, aHRs remained lower in the vaccinated cohort than in the unvaccinated or pre-vaccination cohorts during weeks 5–28 (ratios of aHRs0 0.63 (0.49–0.80) and 0.61 (0.55–0.68), respectively).

The aHRs for VTE were substantially higher during weeks 1–4 after hospitalised COVID-19 (pre-vaccination cohort 88.5 (84.1–93.2), vaccinated cohort 46.7 (42.2–51.6), unvaccinated cohort 199.8 (174.7–228.5)) than after non-hospitalised COVID-19 (pre-vaccination cohort 5.99 (5.61–6.40), vaccinated cohort 2.13 (1.91–2.38), unvaccinated cohort 7.64 (6.34–9.20)). The incidence of VTE was still markedly elevated during weeks 5–28 after hospitalised COVID-19 in the pre-vaccination, vaccinated and unvaccinated cohorts (aHRs 4.39 (4.01–4.80), 5.67 (4.63–6.94) and 7.50 (5.07–11.1)

respectively). In sensitivity analyses restricted to primary diagnosis of VTE, aHRs after COVID-19 diagnosis were attenuated compared with aHRs for all VTEs (Figure S2). This attenuation was particularly marked during weeks 1–4 (including both day 0 and the rest of that period) and after hospitalised COVID-19. In additional analyses splitting follow-up during weeks 1–4 into shorter time intervals, hazard ratios for VTE were generally similar during days 1–6 and days 7–13 after hospitalised COVID-19, then declined during days 14–20 and days 21–27 (Table S6). Hazard ratios after non-hospitalised COVID-19 did not markedly decline between days 1-6 and days 21–27.

In each cohort, aHRs for acute MI during weeks 1–4 after COVID-19 diagnosis, versus before or without COVID-19 diagnosis, were similar to those for ischaemic stroke (Fig. 3, Table 3). In the pre-vaccination cohort, aHRs for acute MI remained elevated during weeks 29–52 (1.16 (1.09–1.24)) and weeks 53–102 (1.31 (1.19–1.45)), but the incidence of ischaemic stroke was only slightly elevated from 29 weeks onwards (aHR 1.16 (1.05–1.27) during weeks 53–102). In all cohorts, aHRs during weeks 1–4 were markedly higher for PE (pre-vaccination 31.7 (30.3–33.1)), vaccinated cohort 9.10 (8.36–9.90), unvaccinated cohort 82.8 (72.7–94.3)) than for DVT, and aHRs for PE remained higher than for DVT during weeks 5–28 (Fig. 3, Table 4). By contrast, in the pre-vaccination cohort aHRs for DVT during weeks 29–102 were higher than for PE.

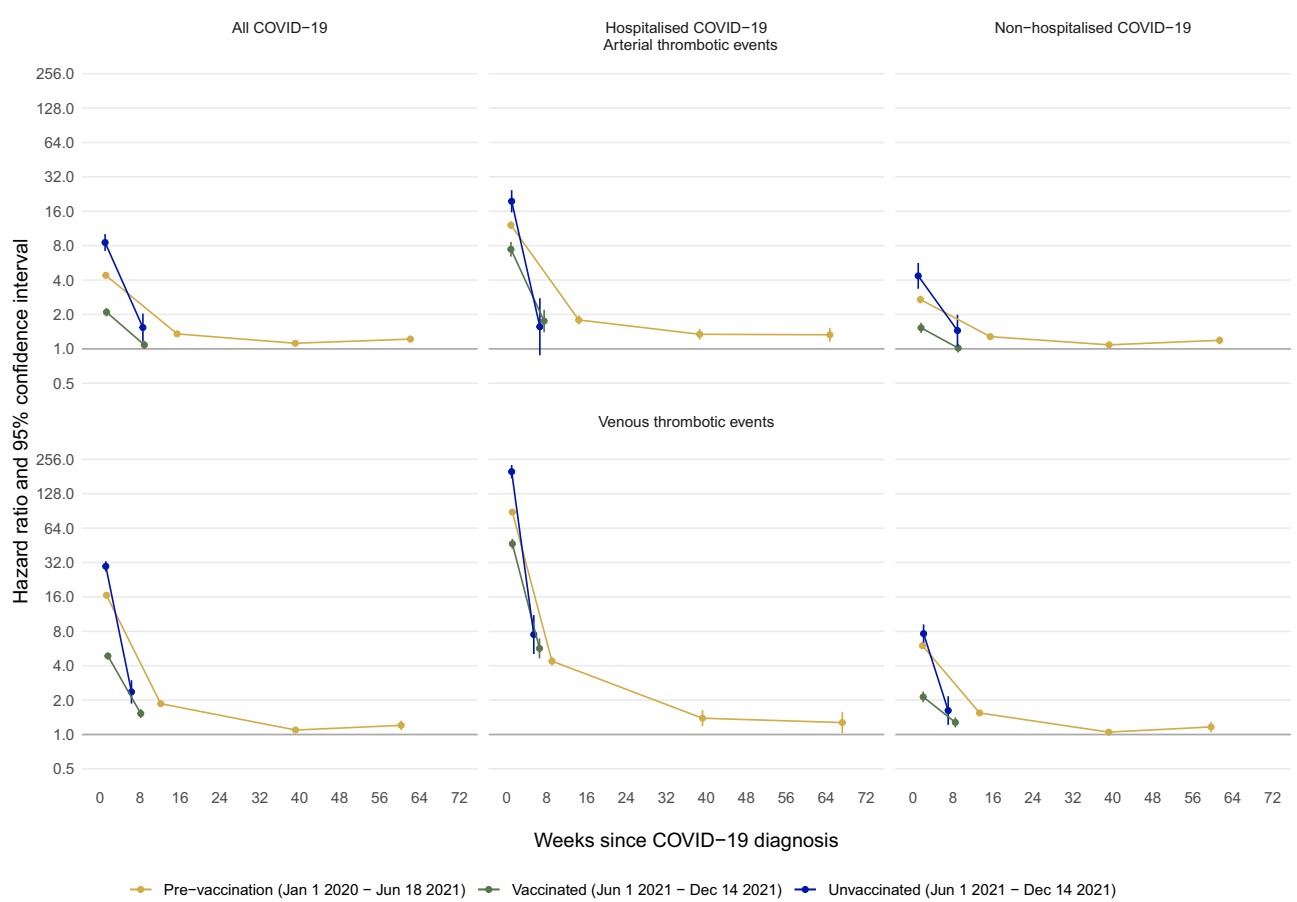

**Fig. 2 | Maximally adjusted hazard ratios and 95% CIs comparing the incidence of arterial thrombotic and venous thrombotic events after versus before or without a COVID-19 diagnosis, in the pre-vaccination, vaccinated and unvaccinated cohorts, overall and by COVID-19 severity.** Upper panels: Maximally adjusted hazard ratios and 95% CIs for arterial thrombotic events. Lower panels: Maximally adjusted hazard ratios and 95% CIs for venous thrombotic events. Left panels: all COVID-19 diagnoses: Middle panels: hospitalised COVID-19. Right panels: non-hospitalised COVID-19. The numbers of people in the pre-vaccination, vaccinated and unvaccinated cohorts were 18,210,937; 13,572,399 and 3,161,485 respectively. The numbers of COVID-19 diagnoses were 1,150,299 (75,667 hospitalised) in the pre-vaccination cohort, 844,235 (15,342 hospitalised) in the vaccinated cohort and 162,103 (9250 hospitalised) in the unvaccinated cohort. Maximally adjusted hazard ratios and 95% CIs are plotted at the median time of the outcome event within each follow up period in each cohort. Events on the day of COVID-19 diagnosis (day 0) were excluded. The numerical values of hazard ratios and their 95% CIs are displayed in Tables 3 and 4.

The incidence of heart failure, angina, and subarachnoid haemorrhage and haemorrhagic stroke during weeks 1–4 after COVID-19 diagnosis was substantially elevated in each cohort, versus before or without COVID-19 diagnosis, although aHRs were lower in the vaccinated cohort than the pre-vaccination or unvaccinated cohorts (Fig. 3, Table 5). Compared with these outcomes, the incidence of transient ischaemic attack was less markedly elevated during weeks 1–4. Though greater than 1, aHRs for these four outcomes were markedly lower during weeks 5–28 than weeks 1-4 after COVID-19 diagnosis. In the pre-vaccination cohort, the incidence of heart failure during weeks 53–102 was similar to the incidence before or without COVID-19 diagnosis (aHR 1.04 (0.98–1.11)). The incidence of angina and transient ischaemic attack was slightly elevated (aHRs between 1.10 and 1.16) and remained elevated during weeks 29–102. aHRs for subarachnoid haemorrhage and haemorrhagic stroke were 1.32 (1.14–1.52) during weeks 29–52 and 1.42 (1.15–1.76) during weeks 53–102.

In subgroup analyses, aHRs for both ATE and VTE were generally lower in younger age groups, in females, and in those reporting white ethnicity (Tables S7, S8, Figures S3, S4). Estimated excess risks of ATE 6 months post-COVID-19 diagnosis were 642, 229 and 718 per 100,000 people diagnosed with COVID-19 in the pre-vaccination, vaccinated and unvaccinated cohorts respectively (Fig. 4, Table S9). Corresponding estimated excess risks of VTE were 797, 270, and 1094 per 100,000 people diagnosed with COVID-19, respectively.

## Discussion

These cohort studies of up to -18.2 million people demonstrate that COVID-19 vaccination substantially attenuates the elevated incidence of arterial and venous thrombotic events after COVID-19 diagnosis. This attenuation appears mainly attributable to the reductions in the severity of COVID-19 caused by vaccination. Among people diagnosed with COVID-19 before the availability of vaccination the incidence of ATEs and VTEs remained elevated by around 20% during the second year after COVID-19 diagnosis, and more so after hospitalised COVID-19, compared with the incidence before or without COVID-19 diagnosis. The associations of VTEs, and particularly PE, with hospitalised COVID-19 were stronger than those for ATEs. The elevations in incidence of arterial and venous thrombotic events after COVID-19 diagnosis was of similar magnitude in the pre-vaccination and unvaccinated cohorts, suggesting that changes in the dominant SARS-CoV-2 variant had little impact compared with that of vaccination.

Most previous studies of cardiovascular events following SARS-CoV-2 infections were conducted prior to the delta era and widespread availability of COVID-19 vaccination[1,21–23]. For example, a study of over 6 million patients from the US Department of Veterans Affairs healthcare system, including over 162,000 who had a positive COVID-19 test between March 2020 and January 2021 estimated HRs 30 days to 12 months post COVID-19 for several outcomes we considered, including MI (HR: 1.63, 95% CI: 1.51–1.75), DVT (2.09, 1.94-2.24), and

**Table 5 | Maximally adjusted hazard ratios (95% CI) comparing the incidence of other vascular events after versus before or without a COVID-19 diagnosis, in the pre-vaccination, vaccinated and unvaccinated cohorts, overall and according to COVID-19 severity**

|  | Time since COVID-19 diagnosis | Pre-vaccination cohort | Vaccinated cohort | Unvaccinated cohort |
|---|---|---|---|---|
| Heart failure | Day 0 | 241.7 (234.8–248.8) | 74.4 (70.9–78.1) | 181.0 (157.9–207.5) |
|  | 1–4 weeks | 4.67 (4.45–4.90) | 2.55 (2.41–2.70) | 7.74 (6.56–9.13) |
|  | 5–28 weeks | 1.43 (1.38–1.48) | 1.14 (1.07–1.21) | 1.87 (1.47–2.40) |
|  | 29–52 weeks | 1.09 (1.05–1.14) | – | – |
|  | 53–102 weeks | 1.04 (0.98–1.11) | – | – |
| Angina | Day 0 | 137.2 (131.3–143.3) | 45.5 (42.3–48.9) | 144.5 (120.4–173.5) |
|  | 1–4 weeks | 4.18 (3.96–4.42) | 2.09 (1.95–2.25) | 6.33 (5.15–7.78) |
|  | 5–28 weeks | 1.23 (1.18–1.28) | 1.12 (1.04-1.20) | 1.34 (0.97-1.86) |
|  | 29–52 weeks | 1.11 (1.06–1.16) | – | – |
|  | 53–102 weeks | 1.16 (1.08–1.25) | – | – |
| Transient ischaemic attack | Day 0 | 46.5 (39.1–55.4) | 21.2 (16.4–27.5) | † |
|  | 1–4 weeks | 2.01 (1.69–2.39) | 1.26 (1.00–1.58) | † |
|  | 5–28 weeks | 1.23 (1.12–1.35) | 1.16 (0.98–1.37) | † |
|  | 29–52 weeks | 1.12 (1.01–1.24) | – | – |
|  | 53–102 weeks | 1.10 (0.95-1.29) | – | – |
| Subarachnoid haemorrhage and haemorrhagic stroke | Day 0 | 287.0 (256.0–321.7) | 91.2 (73.7–113.0) | † |
|  | 1–4 weeks | 4.55 (3.82–5.41) | 2.29 (1.74-3.01) | † |
|  | 5–28 weeks | 1.44 (1.27–1.64) | 1.17 (0.89–1.54) | † |
|  | 29–52 weeks | 1.32 (1.14–1.52) | – | – |
|  | 53–102 weeks | 1.42 (1.15–1.76) | – | – |

† Insufficient events for estimation

heart failure (1.72, 1.65–1.80)[23]. Another study using data from the US Department of Veterans Affairs healthcare system found that vaccination against COVID-19 confers partial protection against a range of health outcomes including cardiovascular outcomes[24].

Many pathways have been postulated to explain the increased risk of vascular events after infection[25]. Respiratory infections including SARS-CoV-2 activate immune responses beyond the respiratory epithelium. Influenza leads to inflammation, smooth muscle proliferation and fibrin deposition in atherosclerotic plaques in mice, which may increase risks of MI if replicated in human infection[26]. SARS-CoV-2 infection generally increases inflammatory cytokines, activating endothelium and increasing biomarkers of thrombosis, but probably does not directly infect the vascular epithelium (although this is contested)[27]. Agents that attenuate the inflammatory response (dexamethasone, tocilizumab and baricitinib) reduce mortality from SARS-CoV-2, but it is unclear whether these reduce vascular-specific mortality[28–31]. Cardiovascular sequalae of SARS and MERS are not well defined due to the small number of patients who have been followed up and lack of appropriate control groups, but the evidence points to potentially substantial post-infection cardiovascular risks[32].

Although the risk of thrombotic events is high soon after infection, clinical trials have not demonstrated clear benefit from antithrombotic agents to most people with COVID-19. In RECOVERY, aspirin did not reduce the chance of death at 28 days after randomisation (rate ratio 0.96, 95%CI: 0.89–1.04), but did modestly reduce hospital stay[33]. In ATTACC, ACTIV-4a, and REMAP-CAP, therapeutic anticoagulation with heparins did not clearly increase organ-support free days in patients who were hospitalised with COVID-19 and were critically unwell (OR: 0.83, CrInt 0.67–1.03), but did in patients who were not critically unwell (4% (CrInt 0.5–7.2))[34,35]. Oral anticoagulation after admission did not clearly reduce mortality in people after hospital admission in HEAL-COVID (n = 402)[36] and ACTION (n = 615)[37], but there was a suggestion of benefit (and no evidence of bleeding) in MICHELLE (n = 320)[38].

We analysed comprehensive linked electronic health record data on ~40% of the English population. The large dataset size allowed us to estimate associations in important subgroups as well as overall, and we controlled for a substantial range of potential confounding factors. Limitations of our study include first that we did not have access to SARS-CoV-2 sequencing data so could not confirm the SARS-CoV-2 variant for individual infections. Second, we could not analyse an Omicron-era cohort as mandatory testing for SARS-CoV-2 in England stopped at the end of March 2022. Third, a substantial proportion of cardiovascular events were recorded on the day of COVID-19 diagnosis (day 0), so it is possible that some COVID-19 diagnoses were made because patients with cardiovascular events were examined in hospitals or in other healthcare settings, rather than because COVID-19 caused these cardiovascular events. However, we separated day 0 from weeks 1-4 and the differences between aHRs in the vaccinated and unvaccinated cohorts during weeks 1-4 suggest that reverse causation was not the main reason for the high hazard ratios. Fourth, although we adjusted for a broad range of potential confounders, our results may be biased by unmeasured confounding due to missing information or measurement error. For example, body mass index is not systematically recorded in electronic health records and may be incorrect in some instances. In addition, we did not adjust for potential time-varying confounders that varied after the cohort start date and may have predicted both COVID-19 and cardiovascular outcomes. Fifth, we did not account for reinfections, although these will have been rare until the Omicron variant became dominant. Sixth, older people in the vaccinated cohort were eligible for booster vaccination from September 2021 so associations between COVID-19 and CVD in that cohort will relate to the effects of the booster as well as the second vaccination. Seventh, outcome misclassification or delays in diagnosis can occur because individuals with mild symptoms do not present to healthcare or because it is difficult to identify a thrombotic disorder in individuals who are very unwell with COVID-19. Eighth, estimated hazard ratios were broadly similar for the pre-vaccination and unvaccinated cohorts. These may have been influenced, beyond the different SARS-CoV-2 variants studied in the two cohorts, by factors such as evolving testing strategies over time, the effect of lockdowns and

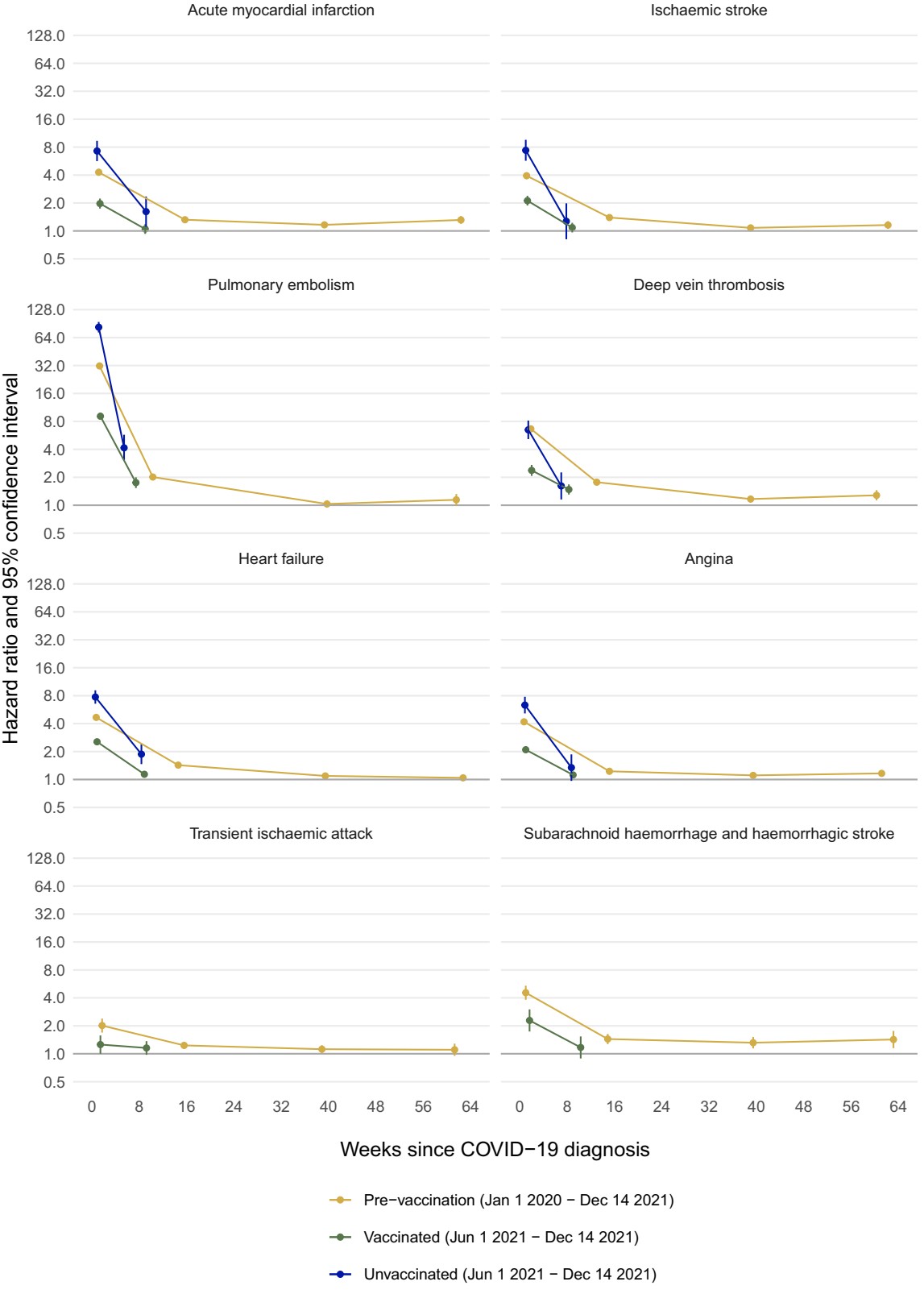

changing access to healthcare services during the pandemic. Finally, we studied the incidence of cardiovascular events after diagnosis of COVID-19. Some COVID-19 cases will not have been recorded in electronic health records for reasons including lack of availability of mass testing in the first phase of the pandemic in 2020 and lack of testing of asymptomatic individuals. The inclusion of some follow-up after COVID-19 in the comparison group is likely to have led to the underestimation of the elevation in the incidence of cardiovascular events after COVID-19.

The absence of clear benefit for antithrombotic strategies after infection, and the clear reduction in thrombotic complications after vaccination means that vaccination is critical to prevent severe COVID-19 disease and its cardiovascular sequelae, particularly in population groups at the highest risk and in which coverage is currently low. In

**Fig. 3 | Maximally adjusted hazard ratios and 95% CIs comparing the incidence of arterial thrombotic, venous thrombotic, and other vascular events after versus before or without a COVID-19 diagnosis, in the pre-vaccination, vaccinated and unvaccinated cohorts.** Upper left panel: Acute myocardial infarction. Upper right panel: Ischaemic stroke. Second row left panel: Pulmonary embolism. Second-row right panel: Deep vein thrombosis. Third row left panel: Heart failure. Third row right panel: Angina. Lower left panel: Transient ischaemic attack. Lower right panel: Subarachnoid haemorrhage and haemorrhagic stroke. The numbers of

people in the pre-vaccination, vaccinated and unvaccinated cohorts were 18,210,937; 13,572,399 and 3,161,485, respectively. The numbers of COVID-19 diagnoses was 1,150,299 in the pre-vaccination cohort, 844,235 in the vaccinated cohort and 162,103 in the unvaccinated cohort. Maximally adjusted hazard ratios and 95% CIs are plotted at the median time of the outcome event within each follow-up period in each cohort. Events on the day of COVID-19 diagnosis (day 0) were excluded. The numerical values of hazard ratios and 95% CIs are displayed in Tables 3, 4 and 5.

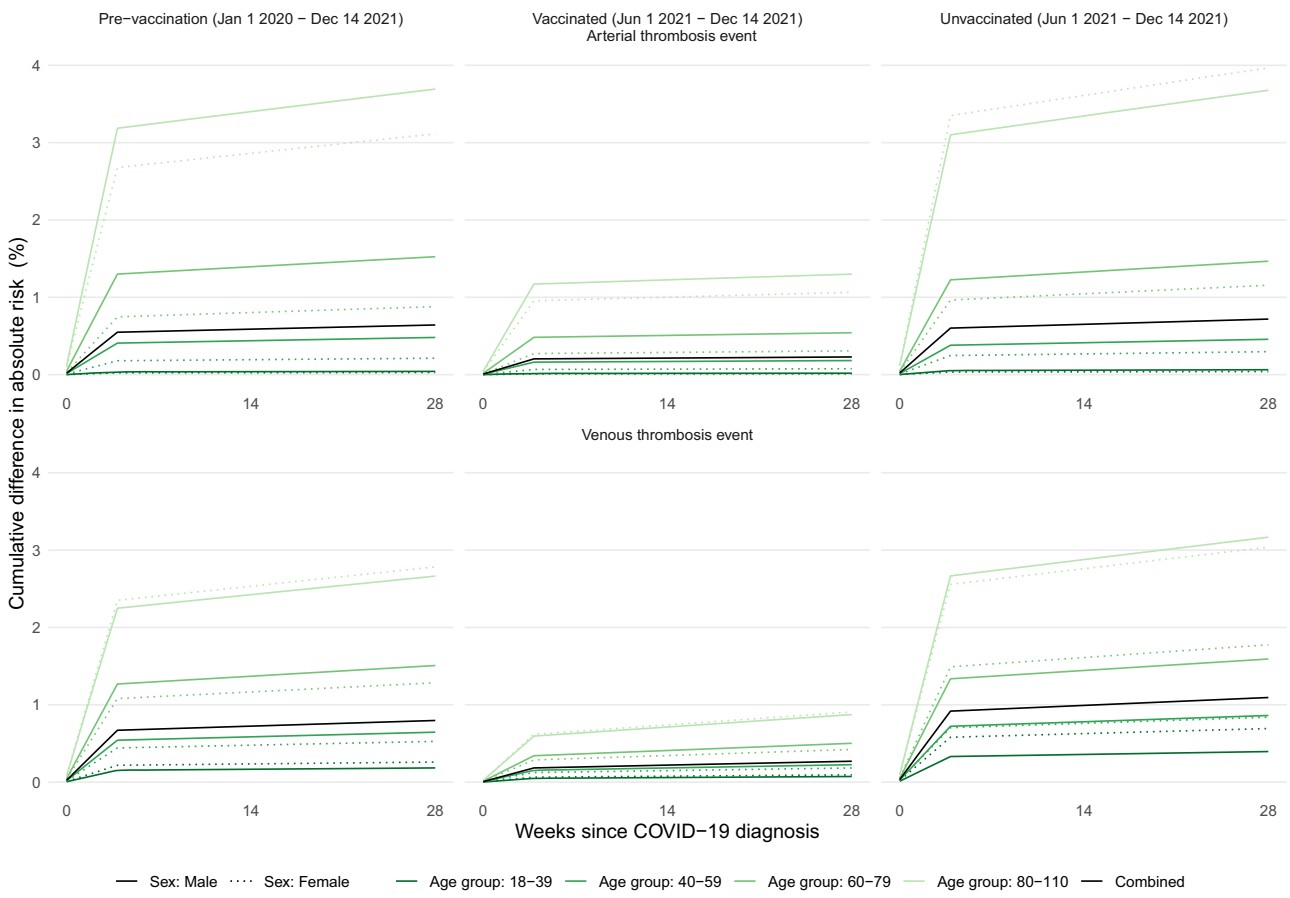

**Fig. 4 | Estimated absolute increase in the risk of arterial thrombotic and venous thrombotic events over time since COVID-19 diagnosis, compared with no COVID-19 diagnosis, in the pre-vaccination, vaccinated and unvaccinated cohorts.** Upper panels: Estimated absolute increase in risk for arterial thrombotic events. Lower panels: Estimated absolute increase in risk for venous thrombotic events. Left panels: pre-vaccination cohort. Middle panels: vaccinated cohort. Right panels: unvaccinated cohort. The numbers of people in the pre-vaccination,

vaccinated and unvaccinated cohorts were 18,210,937; 13,572,399 and 3,161,485 respectively. The number of COVID-19 diagnoses was 1,150,299 in the pre-vaccination cohort, 844,235 in the vaccinated cohort and 162,103 in the unvaccinated cohort. Increases in risks were estimated within sex and age groups, and the estimated overall increase in risk is the average of these (shown in black), weighted according to the proportions in each sex and age group in the pre-vaccination cohort. Estimated excess events at 28 weeks are displayed in Table S9.

conclusion, COVID-19 vaccination reduces the risk of arterial thrombotic, venous thrombotic and other cardiovascular events after COVID-19 diagnosis. People who had COVID-19 before being vaccinated are at higher risk of these events for an extended period of at least two years: further follow-up is required to establish the duration of this higher risk.

## Methods

The study design and pre-determined methods are described in detail in a publicly available protocol: https://github.com/opensafely/post-covid-vaccinated/tree/main/protocol.

### Study design and data source

Using a cohort study design, linked electronic health records were analysed through OpenSAFELY (https://opensafely.org), a data analytics platform created to address COVID-19-related research

questions[39]. The platform provides secure access to primary care records managed by the GP software provider, The Phoenix Partnership (TPP) SystmOne software, which covers around 40% of the population in England. This OpenSAFELY-TPP population is representative of the general population of England in terms of age, sex, ethnicity, index of multiple deprivation, and causes of death[40]. The TPP primary care data is securely linked at the individual level to the Second Generation Surveillance System for Pillars 1 and 2 SARS-COV-2 infection laboratory testing data, COVID-19 vaccination records (National Immunisation Management System), National Health Service (NHS) hospitalisations (Secondary Uses Services data) and the Office of National Statistics (ONS) death registry, including causes of death.

Outcomes were derived from primary care records, hospital admissions and causes of death. Clinically verified SNOMED-CT (Systematized Nomenclature of Medicine-Clinical Terms) and ICD-10

(International Classification of Disease, 10[th] revision) rule-based phenotyping algorithms were used to define arterial thrombosis outcomes (acute MI, ischaemic stroke and a composite 'arterial thrombotic event' (ATE)), venous thrombosis outcomes (PE, DVT and a composite 'venous thrombotic event' (VTE)), and other cardiovascular outcomes (heart failure, angina, transient ischaemic attack, subarachnoid haemorrhage and haemorrhagic stroke). Codes in any position (main or underlying diagnosis or cause of death) were used (see protocol: https://github.com/opensafely/post-covid-vaccinated/tree/main/protocol).

The date of COVID-19 diagnosis was defined as the earliest of the date of a positive SARS-COV-2 test, the date of a confirmed COVID-19 diagnosis in primary or secondary care or the date of death with SARS-COV-2 infection listed as primary or underlying cause. Individuals who were hospitalised with a COVID-19 primary diagnosis within 28 days of first COVID-19 diagnosis were categorised as 'hospitalised COVID-19', otherwise they were categorised as 'non-hospitalised COVID-19'. Covariates identified as potential confounders included age; sex; ethnicity; region; area socioeconomic deprivation; smoking status; number of GP-patient interactions in the last 12 months; and previous history of a specific comorbidity (binary) for a range of diseases (details in Table S10).

## Study population
Three cohorts were defined. In the 'pre-vaccination' cohort, follow-up started on January 1st 2020 (baseline) and ended on the earliest of December 14th 2021 (when the Omicron variant became dominant in England[41], Fig. 1), the date that the outcome event of interest was recorded, and date of death. Exposure was defined as a recorded COVID-19 diagnosis between baseline and the earliest of eligibility for COVID-19 vaccination, date of first vaccination and June 18th 2021 (when all adults became eligible for vaccination): this exposure period was before the Delta variant became dominant in England. The other two cohorts were followed during the period when the Delta variant was dominant in England (Fig. 1): between June 1[st] 2021 (baseline) and December 14th 2021 (study end date). Follow-up in the 'vaccinated' cohort started at the later of baseline and two weeks after a second COVID-19 vaccination and ended at the earliest of the study end date, outcome event date, and date of death. The 'unvaccinated' cohort had not received a COVID-19 vaccine by 12 weeks after they became eligible for vaccination. Follow-up started at the later of baseline and 12 weeks after vaccination eligibility and ended at the earliest of the study end date, outcome event date, date of death, and date of first vaccination.

Individuals eligible for each cohort had been registered with an English GP for at least six months before the cohort baseline, were alive and aged between 18 and 110 years at baseline, and had known sex, region and area deprivation. Individuals with a history of COVID-19 before the cohort baseline were excluded. For each analysis, individuals with a prior history of the outcome analysed were excluded: these individuals were included in one of the subgroup analyses. In the vaccinated cohort, individuals who received a COVID-19 vaccination before December 8[th] 2020, or a second dose before or less than three weeks after their first dose, or received more than one type of vaccine before May 7th 2021, were excluded. In the unvaccinated cohort, individuals who could not be assigned to a vaccination priority group as defined by JCVI were excluded.

## Statistical analyses
For each cohort, baseline demographic and clinical characteristics were described. The number of events per outcome, person-years of follow-up and incidence rates (per 100,000 person-years) of events before and after all, hospitalised and non-hospitalised COVID-19 were calculated. We tabulated the distribution of the number of days between COVID-19 diagnosis and subsequent COVID-19 hospitalisation. Time to first event was analysed for each outcome. Cox models were fitted with calendar time scale using the cohort-specific baseline as the origin (time zero). Hazard ratios (HRs) for follow-up after versus before or without a COVID-19 diagnosis were estimated, splitting follow-up into the day of COVID-19 diagnosis ('day 0'), the remainder of 1-4 weeks and 5-28 weeks after COVID-19 diagnosis for all cohorts, and additionally 29-52 and 53-102 weeks after COVID-19 diagnosis for the pre-vaccination cohort. Because of limited numbers of events in some groups, in subgroup and sensitivity analyses, we estimated hazard ratios by combining day 0 with the rest of weeks 1–4. We additionally estimated hazard ratios 0, 1–6, 7–13, 14–20 and 21–27 days after COVID-19 diagnosis. For each outcome and cohort, we estimated: (i) age and sex; and (ii) maximally (including all potential confounders) adjusted HRs. We calculated ratios of hazard ratios (with corresponding 95% CI) comparing the maximally adjusted HRs for ATE and VTE during weeks 1-4 and 5-28 between the three cohorts. Subgroup analyses were conducted according to whether individuals had been hospitalised for COVID-19 within 28 days of COVID-19 diagnosis. In each cohort absolute excess risks (AER) of any ATE and any VTE after COVID-19, weighted by the proportions of individuals in age and sex strata in the pre-vaccination cohort, were derived based on the maximally adjusted hazard ratios for the cohort. Further details of the statistical analyses and AER calculations are provided in the supplementary material.

For the outcomes ATE and VTE, we conducted additional subgroup analyses by age group, sex, ethnicity, and prior history of the outcome. Further sensitivity analyses included removing censoring at first vaccination in the unvaccinated cohort and identifying outcomes using primary position only (main diagnosis or cause of death).

Data management and analyses were conducted in Python version 3.8.10 and R version 4.0.2.

## Ethical approval and information governance
This study was approved by the Health Research Authority [REC reference 22/PR/0095] and by the University of Bristol's Faculty of Health Sciences Ethics Committee [reference 117269]. Authors involved in data management/analysis successfully passed information governance training and obtained ONS safe researcher accreditation. NHS England is the data controller of OpenSAFELY-TPP. All outputs underwent disclosure checks and were approved by NHS England. Further details of OpenSAFELY information governance are provided in supplemental methods.

## Reporting summary
Further information on research design is available in the Nature Portfolio Reporting Summary linked to this article.

## Data availability
All data were linked, stored and analysed securely within the Open-SAFELY platform: https://opensafely.org/. Data include pseudonymised data such as coded diagnoses, medications and physiological parameters. No free text data are included. Detailed pseudonymised patient data is potentially re-identifiable and therefore not shared. Data can be analysed through the OpenSAFELY platform subject to appropriate agreement, approvals and training as detailed in supplemental methods.

## Code availability
All code and code lists are shared openly for review and re-use under an MIT open license (pre-vaccination cohort: https://github.com/opensafely/post-covid-Pre-vaccinated-cardiovascular; vaccinated and unvaccinated cohorts: https://github.com/opensafely/post-covid-vaccinated)[42,43].

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

## Acknowledgements

This study was supported by the COVID-19 Longitudinal Health and Wellbeing National Core Study, funded by the UKRI Medical Research Council (MC_PC_20059); the COVID-19 Data and Connectivity National Core Study, funded by the UKRI Medical Research Council; and by the CONVALESCENCE long COVID study, funded by the UK National Institute for Health and Care Research (COVID-LT-009). This work was also supported by Health Data Research UK, which is funded by the UK Medical Research Council, Engineering and Physical Sciences Research Council, Economic and Social Research Council, Department of Health and Social Care (England), Chief Scientist Office of the Scottish Government Health and Social Care Directorates, Health and Social Care Research and Development Division (Welsh Government), Public Health Agency (Northern Ireland), British Heart Foundation and Wellcome. GC, SI and AWo are supported by the British Heart Foundation (RG/13/13/30194; RG/18/13/33946), BHF Cambridge Centre of Research Excellence (RE/18/1/34212) and NIHR Cambridge Biomedical Research Centre (BRC-1215-20014; NIHR203312). RK, GDS, and VW are supported by the Medical Research Council Integrative Epidemiology Unit at the University of Bristol [MC_UU_00011/1; MC_UU_00011/4]. RK and JACS are supported by the NIHR Bristol Biomedical Research Centre and by Health Data Research UK South-West. YW was supported by a UKRI MRC Fellowship (MC/W021358/1) and received funding from the UKRI EPSRC Impact Acceleration Account (EP/X525789/1). SI and AWo were funded by a British Heart Foundation–Turing Cardiovascular Data Science Award (BCDSA/100005). SI is funded by the International Alliance for Cancer Early Detection, a partnership among Cancer Research UK C18081/A31373, Canary Center at Stanford University, the University of Cambridge, OHSU Knight Cancer Institute, University College London, and the University of Manchester. RK and JM were supported by NIHR ARC West. RD and JACS were supported by Health Data Research UK. WW is supported by the Chief Scientist's Office (CAF/01/17). AWo and WW are supported by the Stroke Association (SA CV 20/100018). AWo is part of the BigData@Heart Consortium, funded by the Innovative Medicines Initiative-2 Joint Undertaking under grant agreement No 116074. The views expressed are those of the authors and not necessarily those of the NIHR or the Department of Health and Social Care. We are very grateful for all the support received from the OpenSAFELY team and the TPP Technical Operations team throughout this work, and for generous assistance from the information governance and database teams at NHS England and the NHS England Transformation Directorate. We thank the CONVALESCENCE Study Long Covid PPIE group for their input and for sharing their experiences and expertise throughout the duration of the project.

## Author contributions

Author contributions are reported below in line with the Contributor Roles Taxonomy (CRediT). Conceptualization: Rachel Denholm, Angela M Wood, Jonathan A C Sterne, Venexia Walker. Methodology: Rachel Denholm, Angela M Wood, Jonathan A C Sterne, Venexia Walker. Software: Genevieve I Cezard, Rachel Denholm, Rochelle Knight, Yinghui Wei, Lucy Teece, Renin Toms, Harriet J Forbes, Alex J Walker, Louis Fisher, Jon Massey, Lisa E M Hopcroft, Elsie M F Horne, Kurt Taylor, Tom Palmer, Marwa Al Arab, Jose Ignacio Cuitun Coronado, Sam Ip, Simon Davy, Iain Dillingham, Sebastian Bacon, Amir Mehrkar, Catherine Morton, Catherine Hyams, Ben Goldacre, Venexia Walker. Validation: Genevieve I Cezard, Rachel Denholm, Rochelle Knight, Yinghui Wei, Lucy Teece, Renin Toms, Harriet J Forbes, Alex J Walker, Louis Fisher, Jon Massey, Lisa E M Hopcroft, Elsie M F Horne, Kurt Taylor, Tom Palmer, Marwa Al Arab, Jose Ignacio Cuitun Coronado, Sam Ip, Simon Davy, Iain Dillingham, Sebastian Bacon, Amir Mehrkar, Catherine Morton, Catherine Hyams, Ben Goldacre, Venexia Walker. Formal analysis: Genevieve I Cezard, Rachel Denholm, Rochelle Knight, Yinghui Wei, Lucy Teece, Renin Toms, Harriet J Forbes, Kurt Taylor, Marwa Al Arab, Jose Ignacio Cuitun Coronado, Venexia Walker. Investigation: Genevieve I Cezard, Rachel Denholm, Rochelle Knight, Yinghui Wei, Lucy Teece, Renin Toms, Harriet J Forbes, Kurt Taylor, Marwa Al Arab, Jose Ignacio Cuitun Coronado, Venexia Walker. Resources: Alex J Walker, Louis Fisher, Jon Massey, Lisa E M Hopcroft, Simon Davy, Iain Dillingham, Sebastian Bacon, Amir Mehrkar, Catherine Morton, Catherine Hyams, Ben Goldacre. Data curation: Genevieve I Cezard, Rachel Denholm, Rochelle Knight, Yinghui Wei, Lucy Teece, Renin Toms, Harriet J Forbes, Kurt Taylor, Marwa Al Arab, Jose Ignacio Cuitun Coronado, Venexia Walker. Writing - Original Draft: Genevieve I Cezard, William N Whiteley, Jonathan A C Sterne, Venexia Walker. Writing - Review & Editing: Genevieve I Cezard, Rachel Denholm, Rochelle Knight, Yinghui Wei, Lucy Teece, Renin Toms, Harriet J Forbes, Alex J Walker, Louis Fisher, Jon Massey, Lisa E M Hopcroft, Elsie M F Horne, Kurt Taylor, Tom Palmer, Marwa Al Arab, Jose Ignacio Cuitun Coronado, Sam Ip, Simon Davy, Iain Dillingham, Sebastian Bacon, Amir Mehrkar, Catherine Morton, Felix Greaves, Catherine Hyams, George Davey Smith, John Macleod, Nishi Chaturvedi, Ben Goldacre, William N Whiteley, Angela M Wood, Jonathan A C Sterne, Venexia Walker. Visualization: Genevieve I Cezard, Rachel Denholm, Rochelle Knight, Yinghui Wei, Lucy Teece, Renin Toms, Harriet J Forbes, Kurt Taylor, Marwa Al Arab, Jose Ignacio Cuitun Coronado, Venexia Walker. Project administration: Genevieve I Cezard, Rachel Denholm, Angela M Wood, Jonathan A C Sterne, Venexia Walker. Funding acquisition: Nishi Chaturvedi, Angela M Wood, Jonathan A C Sterne.

## Competing interests

WW is supported by the Chief Scientist's Office, the Stroke Association, and the Alzheimer's Society; sits on data monitoring committees for academic trials (TEMPO-2, PROTECT-U, and CATIS-ICAD); and is an independent expert witness to UK courts. NC receives funds from AstraZeneca to support membership of Data Safety and Monitoring Committees for clinical trials. CH is the Principal Investigator of a study which is a collaboration sponsored by the University of Bristol and funded by Pfizer Inc. The other authors report no conflicts.

## Additional information

[1]British Heart Foundation Cardiovascular Epidemiology Unit, Department of Public Health and Primary Care, University of Cambridge, Cambridge, UK. [2]Victor Phillip Dahdaleh Heart and Lung Research Institute, University of Cambridge, Cambridge, UK. [3]Population Health Sciences, University of Bristol, Bristol, UK. [4]NIHR Bristol Biomedical Research Centre, Bristol, UK. [5]Health Data Research UK South-West, Bristol, UK. [6]MRC Integrative Epidemiology Unit, University of Bristol, Bristol, UK. [7]The National Institute for Health and Care Research Applied Research Collaboration West (NIHR ARC West) at University Hospitals Bristol and Weston, Bristol, UK. [8]Centre for Mathematical Sciences, School of Engineering, Computing and Mathematics, University of Plymouth, Plymouth, UK. [9]Department of Population Health Sciences, University of Leicester, Leicester, UK. [10]Population Wellbeing, School of Health Sciences, Cardiff Metropolitan University, Cardiff, UK. [11]Faculty of Epidemiology and Population Health, London School of Hygiene & tropical Medicine, London, UK. [12]The Bennett Institute for Applied Data Science, Nuffield Department of Primary Care Health Sciences, University of Oxford, Oxford, UK. [13]NHS National Services Scotland, Edinburgh, Scotland. [14]Centre for Cancer Genetic Epidemiology, Department of Public Health and Primary Care, University of Cambridge, Cambridge, UK. [15]Digital Environment Research Institute, Queen Mary University of London, London, UK. [16]National Institute for Health and Care Excellence, London, UK. [17]Department of Primary Care and Public Health, Imperial College London, London, UK. [18]MRC Unit for Lifelong Health and Ageing, University College London, London, UK. [19]Centre for Clinical Brain Sciences, University of Edinburgh, Edinburgh, UK. [20]British Heart Foundation Centre of Research Excellence, University of Cambridge, Cambridge, UK. [21]National Institute for Health and Care Research Blood and Transplant Research Unit in Donor Health and Behaviour, University of Cambridge, Cambridge, UK. [22]Health Data Research UK Cambridge, Wellcome Genome Campus and University of Cambridge, Cambridge, UK. [23]Cambridge Centre of Artificial Intelligence in Medicine, Cambridge, UK. [24]Department of Surgery, University of Pennsylvania Perelman School of Medicine, Philadelphia, PA, USA. [25]These authors contributed equally: Genevieve I. Cezard, Rachel E. Denholm, Rochelle Knight, Angela M. Wood, Jonathan A. C. Sterne, Venexia Walker. ✉e-mail: Jonathan.Sterne@bristol.ac.uk

