## [Peer Review File · Nature Communications]

Impact of vaccination on the association of COVID-19 with cardiovascular diseases: an OpenSAFELY cohort studyREVIEWER COMMENTS

Reviewer #1 (Remarks to the Author):

This study looked at the association between COVID-19 and arterial and venous thrombotic diseases. It used the OpenSAFELY dataset, which contains EHR data on about 40% of the population in England. Specifically, the study compared vaccinated and unvaccinated cohorts after vaccines became available and pre-vaccine cohorts before vaccines became available. It divided cohorts into (i) before or no COVID-19 diagnosis, (ii) hospitalized COVID-19, and (iii) non-hospitalized COVID-19. Hazard ratios were adjusted for age, sex, region, ethnicity, area deprivation, smoking status, number of GP-patient interactions, and history of comorbidities. The study found substantial increases in incidence of outcomes during weeks 1-4 after COVID-19, with a much smaller, sustained increase in the following months.

Overall, I think this is an important study that leveraged a unique dataset to understand thrombotic disease risk and COVID-19. However, I have several concerns listed below, especially with the timing of cardiovascular events relative to the date of the positive SARS-COV-2 test. It probably doesn't change the overall finding of thrombotic diseases associated with COVID-19. However, size and duration of increased risk might be overestimated.

Major Comments:

1) Table S4 shows an enormous difference between day 0 and days 1-27 in the "weeks 1-4" cohort. This is only briefly mentioned in the study. However, I think it has a major impact on the findings of this study. By describing the high hazard ratios in "weeks 1-4" and Figures 1 and 2 showing this all the way to almost week 8 (the median of the second time period), the results can be misinterpreted that the high risk of thrombotic diseases lasts for several weeks after initial COVID-19 diagnosis. That is different than significant risk on the day of COVID-19 diagnosis, which then drops very rapidly. Also, without further breakdown in the day 1-27 range, (e.g., day 1, week 1, etc.) it is unclear at what point the risk reached the more stable level seen starting in weeks 5-28.

2) To clarify, in the hospitalized COVID-19 cohort, is Day 0 the date of the earliest positive SARS-COV-2 test or the date when the patient was hospitalized? Assuming it is the positive test date, what is the distribution of number of days after COVID-19 test result when the patient was hospitalized? Did most of the hospitalizations occur on day 0?

3) The authors mention the possibility of "reverse causation", where some of the COVID-19 diagnoses were made because patients who went to a hospital or other healthcare setting with cardiovascular events then got a SARS-COV-2 test incidentally came back positive. They explain that the differences between vaccinated and unvaccinated cohorts suggest that this was not the main reason for the "very high" hazard ratios. I agree with that, but the "very high" hazard ratios were only on day 0. It fell quickly after that. That could be more evidence for reverse causation. Also, patients don't test positive for SARS-COV-2 until a few days after infection and they remain positive for a few days after that. Here, the majority of cardiovascular events occurred on the specific date when the SARS-COV-2 test was performed.

4) Does "without COVID-19" mean (a) without a positive SARS-COV-2 test or (b) with a negative SARS-COV-2 test? These are slightly different. A negative test means that a test was performed for some reason. If the authors had "hospitalized SARS-COV-2 negative" and "non-hospitalized SARS-COV-2 negative" cohorts, it might have provided some insights into potential biases due to selecting patients who had a test recorded in the EHR.

Minor Comments:

5) A figure showing a timeline of the different cohorts, variant types, and vaccine availability and mandate dates would be helpful. It is somewhat difficult to follow this just from the text.

Reviewer #2 (Remarks to the Author):

I have reviewed the manuscript entitled "Impact of vaccination on the association of COVID-19 with arterial and venous thrombotic diseases: an OpenSAFELY cohort study using linked electronic health records."

Firstly, I'd like to acknowledge the significance of the subject matter. The study aptly utilizes a large-scale database from the UK, reflecting the increasing reports emerging from various real-world data. The broad nationwide database that the researchers employed offers delineated results based on infection status and post-infection timelines, with Figure 3 being particularly insightful.

However, I believe certain key modifications would enhance the manuscript:

1. Population Breakdown and Baseline Information: It's challenging to ascertain how the three populations in the study are constructed and what the baseline status of COVID-19 infection is among them. A 'figure' summarizing the populations, exposure, and comparable groups presented in Table 1 would significantly aid readers' comprehension.

2. Concerns Regarding the Incidence Rate in Table 2: I find the presented incidence rate, especially without any adjustments or corrections, may be problematic. In my experience of similar studies and academic writings, unadjusted events and incidences have unfortunately been leveraged by factions strongly opposed to vaccination. Given that the unvaccinated cohort primarily consists of a younger and healthier population, presenting data without age-based baseline adjustments, weighting, or matching could be misleading. If it's not feasible to make additional adjustment for baseline differences, I suggest that the mean age should be shown in Table 1, and age standardization should be attempted for the reported incidence rates(optional).

3. Duration of Observation for Figure 3: While Figure 3 is valuable, it appears the observation period for the population after June 1, 2021, is not sufficiently long. Are there any strategies to mitigate this limitation?

4. Unconfirmed COVID-19 Cases: Even among individuals in the study cohort without recorded COVID-19 infections, various antibody surveys in the UK indicate the presence of a substantial number of undetected infections. How have the authors considered or addressed the potential inclusion of these unconfirmed cases in the cohort?

In conclusion, this research addresses a pivotal area of study. I believe that addressing these concerns and making the suggested modifications will further augment the manuscript's contribution to our understanding of COVID-19, its implications, and the role of vaccination.

Reviewer #3 (Remarks to the Author):

The study by Cezard et al. used data from OpenSAFELY examined the risk of arterial and venous thrombotic diseases in three cohorts: prevaccination, vaccinated and unvaccinated. The dataset used is comprehensive, and the results are presented with multifaceted insights. An increased risk of outcomes following a COVID-19 infection was observed across all three cohorts, and the authors drew indirect inferences about the impact of vaccination on these outcomes. However, I have a major concern about the construction of control group, and have few additional comments.

1. In supplementary method, the authors described "the datasets analysed included all individuals with the outcome event or who had COVID-19 during follow up, and a randomly selected subset of those without either the outcome or COVID-19 ("non-exposed controls")."

This methodology poses a potential issue. By including all non-COVID-19 participants who experienced the outcome and only randomly selecting non-COVID-19 participants without the outcome, there's a risk of introducing a significant bias. Such biases cant not be adequately address through weighting.

This analytical approach artificially elevated the event rate within the non-COVID-19 (control) group, thereby create doubt on the accuracy of the results, especially when evaluating the effects of COVID-19 against this potentially skewed control group.

It would been more appropriate to randomly select participants from the non-COVID-19 group without consideration of their outcome status.

2. The study's title emphasizes the impact of vaccination, yet the abstract and results predominantly highlight the association between COVID-19 and outcomes across the three cohorts. While the authors suggest that comparisons among these cohorts could isolate differing effects, a direct comparison within the COVID-19 patients across these groups seems absent. As a result, only indirect inferences were made on one of the major research question of the study.

2-1. Difference on risk between the pre-vaccination and unvaccinated cohorts might be attributed to factors beyond just changing variants. Over time, improved healthcare practices, a deeper understanding of COVID-19, and evolving strategies may play roles. Additionally, the effects of lockdowns and limited healthcare visits, especially at the onset of the pandemic, could influence the risk difference.

3. If outcomes for participants were recorded in specialty care settings, how confident are we that the study data captures these events comprehensively?

4. How is the history of the outcome accounted for? Is there any effort to reduce bias from the carryon of diagnosis code from before infection? How is repeat infection during follow up accounted for in the study?

5. In the discussion, the authors reference a study from the US Department of Veterans to highlight the lack of studies in the Delta era. Yet, another study from that same group, which studied breakthrough infection—a topic closely aligned with the current study—has been notably omitted. (PMID: 35614233)

Minor

1. In the "number of events and incidence rate" section, it would be helpful to include percentages alongside raw numbers. Additionally, consider presenting the incidence rate within the main text for clearer communication.

2. The effort to present risks on an absolute scale (Page 12, lines 226-229) is commendable. However, the AER seems to be standardized only by age and sex. These rates would not be comparable or meaningful if they were not based on consistent population characteristics, particularly when considering potential differences in the risk factors for outcomes across the cohorts.

3. In Tables S3 and S6, the time period indicated as weeks "5-8" seems like it should be "5-28". Please verify.

4. The figures representing HRs across weeks could benefit from some enhancements. Currently, the information from different cohorts overlaps, making it challenging to be seen. Additionally, the estimates and confidence intervals for each time point are somewhat obscured due to the y-axis range. Further refinement would aid in clarity and readability.

We thank the editors for inviting us to submit a revised version of our manuscript entitled “Impact of vaccination on the association of COVID-19 with arterial and venous thrombotic diseases: an OpenSAFELY cohort study using linked electronic health records”. We are grateful to the reviewers for their constructive comments, which we think have markedly improved the revised manuscript. We have provided a point-by-point response below with responses presented in italics.

REVIEWER COMMENTS

Reviewer #1 (Remarks to the Author):

This study looked at the association between COVID-19 and arterial and venous thrombotic diseases. It used the OpenSAFELY dataset, which contains EHR data on about 40% of the population in England. Specifically, the study compared vaccinated and unvaccinated cohorts after vaccines became available and pre-vaccine cohorts before vaccines became available. It divided cohorts into (i) before or no COVID-19 diagnosis, (ii) hospitalized COVID-19, and (iii) non-hospitalized COVID-19. Hazard ratios were adjusted for age, sex, region, ethnicity, area deprivation, smoking status, number of GP-patient interactions, and history of comorbidities. The study found substantial increases in incidence of outcomes during weeks 1-4 after COVID-19, with a much smaller, sustained increase in the following months.

Overall, I think this is an important study that leveraged a unique dataset to understand thrombotic disease risk and COVID-19. However, I have several concerns listed below, especially with the timing of cardiovascular events relative to the date of the positive SARS-COV-2 test. It probably doesn't change the overall finding of thrombotic diseases associated with COVID-19. However, size and duration of increased risk might be overestimated.

Thank you for your careful review. We address these points in detail below.

Major Comments:

1) Table S4 shows an enormous difference between day 0 and days 1-27 in the “weeks 1-4” cohort. This is only briefly mentioned in the study. However, I think it has a major impact on the findings of this study. By describing the high hazard ratios in “weeks 1-4” and Figures 1 and 2 showing this all the way to almost week 8 (the median of the second time period), the results can be misinterpreted that the high risk of thrombotic diseases lasts for several weeks after initial COVID-19 diagnosis. That is different than significant risk on the day of COVID-19 diagnosis, which then drops very rapidly.

As noted by the reviewer, Supplementary table 4 in our original manuscript showed the results of a sensitivity analysis where we separated hazard ratios on day 0 from the rest of the 1-4 weeks period. This table showed substantial differences between the hazard ratios on day 0 and days 1-27. We were careful to describe these results in our original manuscript. However, we recognise that the substantial differences between the day 0 hazard ratios and those for the rest of weeks 1-4 may lead to the misinterpretation than concerns the reviewer. Therefore, we have amended the main analyses to report the day 0 time period separately. The text of the results section, table 3, and supplementary table 2 have been amended accordingly. Figures 1 and 2 now exclude events and hazard ratios on day 0.

In the methods section, page 21, lines 412-417, we now write:

“Hazard ratios (HRs) for follow-up after versus before or without a COVID-19 diagnosis were estimated, splitting follow-up into the day of COVID-19 diagnosis (‘day 0’), the remainder of 1-4 weeks and 5-28 weeks after COVID-19 diagnosis for all cohorts, and additionally 29-52 and 53-102 weeks after COVID-19 diagnosis for the pre-vaccination cohort. Because of limited numbers of events in some groups, in subgroup and sensitivity analyses we estimated hazard ratios combining day 0 with the rest of weeks 1-4.”

On page 9, lines 164-168 as follows:

“For all outcomes, hazard ratios were extremely high on the day of COVID-19 diagnosis (day 0) particularly among individuals hospitalised with COVID-19 on the day of diagnosis. The incidence of each outcome in each cohort was also elevated during weeks 1-4 after COVID-19 diagnosis, compared with before or without COVID-19 diagnosis.”

Also, without further breakdown in the day 1-27 range, (e.g., day 1, week 1, etc.) it is unclear at what point the risk reached the more stable level seen starting in weeks 5-28.

We have added a new supplementary Table S4, which displays hazard ratios for all cohorts into 0, 1-6, 7-13, 14-20 and 21-27 days after COVID-19 diagnosis.

In the methods section (page 21 lines 417-419) we now write: “We additionally estimated hazard ratios 0, 1-6, 7-13, 14-20 and 21-27 days after COVID-19 diagnosis.”

In the results section (page 10 lines 190-192) we now write: “In additional analyses splitting follow-up during weeks 1-4 into shorter time intervals, hazard ratios for ATE declined steadily from days 1-6 to days 21-27 after COVID-19 diagnosis, in all cohorts (Table S4)” and on pages 11-12 (lines 213-216) we now write: “In additional analyses splitting follow-up during weeks 1-4 into shorter time intervals, hazard ratios for VTE were generally similar during days 1-6 and days 7-13 after hospitalised COVID-19, then declined during days 14-20 and days 21-27 (Table S4). Hazard ratios after non-hospitalised COVID-19 did not markedly decline between days 1-6 and days 21-27”.

2) To clarify, in the hospitalized COVID-19 cohort, is Day 0 the date of the earliest positive SARS-COV-2 test or the date when the patient was hospitalized? Assuming it is the positive test date, what is the distribution of number of days after COVID-19 test result when the patient was hospitalized? Did most of the hospitalizations occur on day 0?

Yes, for hospitalised COVID-19, day 0 is the day of COVID-19 diagnosis. We added a new supplementary table, Table S9, which shows the distribution of number of days between COVID-19 diagnosis and hospitalisation.

In the methods section on page 21, lines 409-410, we now write:

“We tabulated the distribution of number of days between COVID-19 diagnosis and subsequent COVID-19 hospitalisation.”

In the results section on page 8, lines 126-129, we now write:

“Among 121,514 people in the pre-vaccination cohort who were hospitalised because of COVID-19 62,163 (51.2%) were hospitalised on the day of diagnosis and 11,413 (9.4%) were hospitalised the day after diagnosis: the remainder were hospitalised 2-28 days after diagnosis (Table S9).”

3) The authors mention the possibility of “reverse causation”, where some of the COVID-19 diagnoses were made because patients who went to a hospital or other healthcare setting with cardiovascular events then got a SARS-COV-2 test incidentally came back positive. They explain that the differences between vaccinated and unvaccinated cohorts suggest that this was not the main reason for the “very high” hazard ratios. I agree with that, but the “very high” hazard ratios were only on day 0. It fell quickly after that. That could be more evidence for reverse causation. Also, patients don’t test positive for SARS-COV-2 until a few days after infection and they remain positive for a few days after that. Here, the majority of cardiovascular events occurred on the specific date when the SARS-COV-2 test was performed.

We agree that the date of positive SARS-CoV-2 test is likely to be some days after infection for most people. Cardiovascular events that occurred before positive test or clinical diagnosis will have been classified as occurring before or without a COVID-19 diagnosis, so included in the comparison group. This may have led to some attenuation of estimated hazard ratios.

As discussed above, we now separate the main results into hazard ratios on day 0 and hazard ratios during the rest of weeks 1-4. In the pre-vaccination and unvaccinated cohorts, hazard ratios were substantial during weeks 1-4 even after excluding day 0. The consistently lower hazard ratios in the vaccinated cohort, compared with the pre-vaccination and unvaccinated cohorts, is important evidence against reverse causation being the main reason for the very high hazard ratios on day 0 and during weeks 1-4. We have retained the text in the discussion section making this point (page 16 lines 307-309 in the revised manuscript):

“However, the substantial differences between aHRs in the vaccinated and unvaccinated cohorts suggest that such reverse causation was not the main reason for the very high hazard ratios during weeks 1-4.”

4) Does “without COVID-19” mean (a) without a positive SARS-COV-2 test or (b) with a negative SARS-COV-2 test? These are slightly different. A negative test means that a test was performed for some reason. If the authors had “hospitalized SARS-COV-2 negative” and “non-hospitalized SARS-COV-2 negative” cohorts, it might have provided some insights into potential biases due to selecting patients who had a test recorded in the EHR.

Our analysis does not include negative tests. Therefore, “without COVID-19” refers to the absence of a COVID-19 diagnosis. Diagnosis is identified through positive test as well as from primary care, secondary care and death data sources (using the relevant COVID-19 specific SNOMED and ICD-10 codes). This is specified on page 19, lines 367-369 as follows:

“The date of COVID-19 diagnosis was defined as the earliest of the date of a positive SARS-COV-2 test, the date of a confirmed COVID-19 diagnosis in primary or secondary care or the date of death with SARS-COV-2 infection listed as primary or underlying cause.”

We added the word “diagnosis” in the following sentence (page 21, lines 412-416) to provide a better hint that “without COVID-19 diagnosis” means in the absence of a COVID-19 diagnosis (not necessarily based on testing solely):

“Hazard ratios (HRs) for follow-up after versus before or without a COVID-19 diagnosis were estimated, splitting follow-up into the day of COVID-19 diagnosis (‘day 0’), the remainder of 1-4 weeks and 5-28 weeks after COVID-19 diagnosis for all cohorts, and additionally 29-52 and 53-102 weeks after COVID-19 diagnosis for the pre-vaccination cohort.”

To ensure that the meaning of “without” is clear, the results section has been updated throughout to use “without COVID-19 diagnosis” rather than “without COVID-19”. Some of the discussion section and the titles of tables and figures has also been updated when relevant.

Minor Comments:

5) A figure showing a timeline of the different cohorts, variant types, and vaccine availability and mandate dates would be helpful. It is somewhat difficult to follow this just from the text.

We agree, and have included a new Figure S6 in the supplementary material. This displays the estimated number of COVID-19 cases identified by community testing in England between January 2020 and January 2022, and shows the start and end of follow-up for each cohort, the periods during which different variants were dominant and the times that mass testing became available and vaccination rollout commenced. Figure S6 provides visual support to the definition of each cohort: it is now referenced in the manuscript page 19 line 380 as follows:

“In the ‘pre-vaccination’ cohort, follow-up started on January 1st 2020 (baseline) and ended on the earliest of December 14th 2021 (when the Omicron variant became dominant in England⁴⁰, Figure S6)”

and page 20 line 385 as follows:

“The other two cohorts were followed during the period when the Delta variant was dominant in England (Figure S6)”.

We suggest to the editors that, if space permits, it would be helpful to include Figure S6 as a main figure in the paper.

Reviewer #2 (Remarks to the Author):

I have reviewed the manuscript entitled "Impact of vaccination on the association of COVID-19 with arterial and venous thrombotic diseases: an OpenSAFELY cohort study using linked electronic health records."

Firstly, I'd like to acknowledge the significance of the subject matter. The study aptly utilizes a large-scale database from the UK, reflecting the increasing reports emerging from various real-world data. The broad nationwide database that the researchers employed offers delineated results based on infection status and post-infection timelines, with Figure 3 being particularly insightful.

However, I believe certain key modifications would enhance the manuscript:

1. Population Breakdown and Baseline Information: It's challenging to ascertain how the three populations in the study are constructed and what the baseline status of COVID-19 infection is among them. A 'figure' summarizing the populations, exposure, and comparable groups presented in Table 1 would significantly aid readers' comprehension.

We agree, as noted in our response to a similar comment by Reviewer 1 above, and have now included a new Figure S6 which depicts the requested information.

Individuals with a diagnosis of COVID-19 before the start of follow-up in each cohort were excluded. We note this in our original manuscript and in the revised manuscript on page 20, pages 396-397:

“Individuals with a history of COVID-19 before the cohort baseline were excluded.”

2. Concerns Regarding the Incidence Rate in Table 2: I find the presented incidence rate, especially without any adjustments or corrections, may be problematic. In my experience of similar studies and academic writings, unadjusted events and incidences have unfortunately been leveraged by factions strongly opposed to vaccination. Given that the unvaccinated cohort primarily consists of a younger and healthier population, presenting data without age-based baseline adjustments, weighting, or matching could be misleading. If it's not feasible to make additional adjustment for baseline differences, I suggest that the mean age should be shown in Table 1, and age standardization should be attempted for the reported incidence rates(optional).

We agree that the crude incidence rates presented in Table 2 could be misinterpreted in the way that the reviewer suggests, and explained the age difference in our original manuscript. We do not wish to present adjusted or standardized incidence rates in Table 2, because this aims to present the raw data on which subsequent adjusted analyses (presented in Table 3 and Figures 1 and 2) are based. To further protect against misinterpretation, we have made the following changes to emphasise the age differences between cohorts in our revised manuscript.

1- *We have described the median age of each cohort on page 8, lines 129-130 and lines 132-134 as follows:*

“In the pre-vaccination cohort, the median age was 49 years (interquartile range (IQR) 34-64)”

“The median (IQR) age was 54 (IQR 39-68) years in the vaccinated cohort, compared with 36 (IQR 28-47) years in the unvaccinated cohort”

2- *We have now added median and IQR age in table 1.*

3- *In the header for Table 2, we now note the median age in each cohort.*

4- *In our original manuscript, we explained that there were generally lower incidence rates in the younger unvaccinated group. In addition, we have modified this to emphasize the age differences across cohort on page 9 lines 149-152 as follows:*

“Incidence rates were generally lower in the unvaccinated cohort than in the vaccinated cohort, as expected given that the median age of the unvaccinated cohort (36 years) was much lower than that of the vaccinated cohort (54 years).”

3. Duration of Observation for Figure 3: While Figure 3 is valuable, it appears the observation period for the population after June 1, 2021, is not sufficiently long. Are there any strategies to mitigate this limitation?

There is no figure 3 in our manuscript: we think that the reviewer was referring to figures 1 and 2.

The end of follow-up for our study (December 14th 2021) was determined by the start of dominance of the Omicron variant and the end of mandatory free testing for SARS-CoV-2. Inevitably, the length of follow-up is longer for the pre-vaccination cohort than for the vaccinated and unvaccinated cohorts.

We explain in the discussion section (page 16, lines 301-303) that “we could not analyse an Omicron-era cohort as mandatory testing for SARS-CoV-2 in England stopped at the end of March 2022.”

4. Unconfirmed COVID-19 Cases: Even among individuals in the study cohort without recorded COVID-19 infections, various antibody surveys in the UK indicate the presence of a substantial number of undetected infections. How have the authors considered or addressed the potential inclusion of these unconfirmed cases in the cohort?

We agree that not all SARS-CoV-2 infections were recorded in electronic health records, for reasons including lack of availability of mass testing in the first phase of the pandemic and lack of testing of asymptomatic individuals. Therefore, some individuals with unidentified COVID-19 will have been included in the comparison group (before or without a COVID-19 diagnosis) in our analyses, which will have led to underestimation of the elevation in incidence of cardiovascular events after COVID-19 diagnosis. We now address this limitation in the discussion as follows (pages 17, lines 324-330):

“Finally, we studied the incidence of cardiovascular events after diagnosis of COVID-19. Some COVID-19 cases will not have been recorded in electronic health records for reasons including lack of availability of mass testing in the first phase of the pandemic in 2020 and lack of testing of asymptomatic individuals. Inclusion of some follow-up after COVID-19 in the comparison group is likely to have led to underestimation of the elevation in incidence of cardiovascular events after COVID-19.”

We now also make clear throughout the manuscript that our findings refer to the incidence of cardiovascular events after COVID-19 diagnosis rather than after COVID-19.

In conclusion, this research addresses a pivotal area of study. I believe that addressing these concerns and making the suggested modifications will further augment the manuscript's contribution to our understanding of COVID-19, its implications, and the role of vaccination.

Thank you for your positive comments on our manuscript.

Reviewer #3 (Remarks to the Author):

The study by Cezard et al. used data from OpenSAFELY examined the risk of arterial and venous

thrombotic diseases in three cohorts: pre-vaccination, vaccinated and unvaccinated. The dataset used is comprehensive, and the results are presented with multifaceted insights. An increased risk of outcomes following a COVID-19 infection was observed across all three cohorts, and the authors drew indirect inferences about the impact of vaccination on these outcomes. However, I have a major concern about the construction of control group, and have few additional comments.

1. In supplementary method, the authors described “the datasets analysed included all individuals with the outcome event or who had COVID-19 during follow up, and a randomly selected subset of those without either the outcome or COVID-19 (“non-exposed controls”).”

This methodology poses a potential issue. By including all non-COVID-19 participants who experienced the outcome and only randomly selecting non-COVID-19 participants without the outcome, there's a risk of introducing a significant bias. Such biases can't not be adequately address through weighting.

This analytical approach artificially elevated the event rate within the non-COVID-19 (control) group, thereby create doubt on the accuracy of the results, especially when evaluating the effects of COVID-19 against this potentially skewed control group.

It would been more appropriate to randomly select participants from the non-COVID-19 group without consideration of their outcome status.

Thank you for the opportunity to clarify these important issues. We are confident that the random sampling that we used to make computations feasible did not introduce bias, and have conducted extensive simulation studies to confirm this.

First, the numbers of events, person-years of follow-up and incidence rates shown in table 2 are based on the full population. Similarly, table 1 is based on the whole dataset. We have clarified in the supplementary methods section that the strategy of random sampling followed by inverse probability weighting was used for the Cox regression analyses.

As explained in the Supplementary methods, the strategy of random sampling followed by inverse probability weighting was required because the very large datasets available within OpenSAFELY meant that some analyses of the whole dataset were not computationally feasible. Because we were sampling from the whole dataset, we were able to record each individual's probability of being sampled. The inverse probability weight (IPW) is simply the reciprocal of this probability: for example if someone was certain to be sampled because they had COVID-19 their IPW is 1. If someone's probability of being sampled was 0.2 (20%) then their IPW is 5. They are then treated as five people in the analysis, and thus represent the similar individuals who were not sampled. Of course, standard errors would be too large if we pretended that their data was really that of five individuals – this problem is solved by using robust standard errors.

Inverse probability (IP) weighting to account for known or estimated sampling probabilities underpins much of survey statistics and is of increasing importance in epidemiology. For example, IP weighting can be used to correct for selection bias due to informative censoring, and is the means through which marginal structural models control time-varying as well as baseline confounding. For a detailed explanation of the approach, we suggest explanations provided in Hernan and Robins' book “What If” – see https://www.hsph.harvard.edu/miguel-hernan/wp-content/uploads/sites/1268/2023/10/hernanrobins_WhatIf_30sep23.pdf. Section 2.4 (starting on page 20) explains the use of IP weighting to control confounding and gives a formal definition of IP

weights. Chapter 12 explains the use of these weights to fit estimate causal effects based on marginal structural models.

We could have randomly sampled from the whole population and conducted unweighted analyses, but this would have reduced precision (there were limited numbers of some events, and in time-to-event analyses the precision of estimated effects is determined by the number of events). It would also have made it more difficult to control confounding, because regression models require that we model the association of the confounders with the outcome so require sufficient outcome events at each level of the confounder. IP weights provide an elegant solution, which can be considered a generalization of the case-control study approach. By randomly sampling people who were not diagnosed with COVID-19 and did not experience the outcome event, and then using IP weighting in the analysis, we can analyse the data as if they came from the whole population and achieve results almost identical to those that would have been found had we been able to fit the model to the whole population. We have confirmed the utility of the approach by conducting extensive simulation studies with varying sampling fractions (e.g. 2%, 5% and 10%). Regardless of the sampling fraction, the results are identical on average. Because there are so many more individuals without than with either diagnosis of COVID-19 or the outcome event, the inflation in the standard errors is minimal.

2. The study's title emphasizes the impact of vaccination, yet the abstract and results predominantly highlight the association between COVID-19 and outcomes across the three cohorts. While the authors suggest that comparisons among these cohorts could isolate differing effects, a direct comparison within the COVID-19 patients across these groups seems absent. As a result, only indirect inferences were made on one of the major research question of the study.

Thank you for the opportunity to clarify this important issue. We think that we do make direct comparisons between the cohorts, which can be made by contrasting the hazard ratios across the rows in in Tables 3 and S2 and, correspondingly, through the vertical differences between the hazard ratio plots for each cohort in Figures 1 and 2. We could have made formal comparisons by computing ratios of hazard ratios and corresponding p-values for interaction, but did not do so because most such p-values would be extremely small because of the very large datasets analysed. In these circumstances, in which sampling variation can be excluded as an explanation for between-cohort differences, it is appropriate to focus on more qualitative comparisons: for example that hazard ratios in the vaccinated cohort were markedly lower than those in the pre-vaccination and unvaccinated cohorts, and that between cohort differences are less marked among people hospitalised with COVID-19. Throughout the results section we directly compare hazard ratios between the cohorts.

2-1. Difference on risk between the pre-vaccination and unvaccinated cohorts might be attributed to factors beyond just changing variants. Over time, improved healthcare practices, a deeper understanding of COVID-19, and evolving strategies may play roles. Additionally, the effects of lockdowns and limited healthcare visits, especially at the onset of the pandemic, could influence the risk difference.

Thank you – we agree and have further addressed this point in the discussion section of our revised manuscript (pages 16-17, lines 321-324):

“Eighth, estimated hazard ratios were broadly similar for the pre-vaccination and unvaccinated cohorts. These may have been influenced, beyond the different SARS-CoV-2 variants studied in the two cohorts, by factors such as evolving testing strategies over time, the effect of lockdowns and changing access to health care services during the pandemic.”

3. If outcomes for participants were recorded in specialty care settings, how confident are we that the study data captures these events comprehensively?

Because the outcome events studied are of clear clinical significance, we expect that recording of them within NHS electronic health care records will be complete or near complete. Private health care is only a small proportion of health care in the UK, and private providers do not generally provide emergency care for events such as myocardial or stroke. In summary, we are confident that the study data captures the outcome events comprehensively.

4. How is the history of the outcome accounted for? Is there any effort to reduce bias from the carryon of diagnosis code from before infection? How is repeat infection during follow up accounted for in the study?

Individuals with a previous history of the outcome event were excluded from analyses with that event as the outcome, although they were included in the subgroup analysis comparing hazard ratios in people with and without a prior history of ATE and VTE. This is now specified in the ONLINE METHODS, in the study population section as follows (page 20, lines 397-399):

“For each analysis, individuals with a prior history of the outcome analysed were excluded: these individuals were included in one of the subgroup analyses.”

In each cohort, follow-up ends at the time of the first occurrence of the outcome event. Therefore outcome events before a diagnosis of COVID-19 are included in the comparison group (before or without COVID-19).

In each cohort, all follow-up time after the first COVID-19 diagnosis is included in our analysis. Repeat infection is not accounted for in our analyses, but was in any case comparatively rare.

5. In the discussion, the authors reference a study from the US Department of Veterans to highlight the lack of studies in the Delta era. Yet, another study from that same group, which studied breakthrough infection—a topic closely aligned with the current study—has been notably omitted. (PMID: 35614233)

Thank you for recommending this reference. It has now been added as follows (page 14, lines 270-272):

“Another study using data from the US Department of Veterans Affairs healthcare system found that vaccination against COVID-19 confers partial protection against a range of health outcomes including cardiovascular outcomes²⁴.”

24. Al-Aly Z, Bowe B, Xie Y. Long COVID after breakthrough SARS-CoV-2 infection. *Nat Med.* 2022;28:1461–1467.

Minor

1. In the "number of events and incidence rate" section, it would be helpful to include percentages alongside raw numbers. Additionally, consider presenting the incidence rate within the main text for clearer communication.

We carefully considered including percentages in Table 2 and the corresponding paragraph of the results section of our manuscript, but think that these are difficult to interpret and potentially misleading. This is because follow-up was much longer in the pre-vaccination cohort than in the vaccinated or unvaccinated cohort. The incidence rates presented in Table 2 account for the person-years of follow-up in each cohort. We summarise the pattern of incidence rates in the results section of our manuscript, on pages 8-9 lines 147-152. "In each cohort, the incidence of each arterial thrombotic and venous thrombotic event was higher after COVID-19 than before or without COVID-19. For each outcome and cohort, the highest incidence rates were after hospitalised COVID-19. Incidence rates were generally lower in the unvaccinated cohort than in the vaccinated cohort, as expected given that the median age of the unvaccinated cohort (36 years) was much lower than that of the vaccinated cohort (54 years)."

The remainder of the results section focusses, in our view appropriately, on the estimated hazard ratios. These are adjusted for potential confounders and make direct comparisons between follow-up after a diagnosis of COVID-19 with follow-up before or without a diagnosis of COVID-19.

2. The effort to present risks on an absolute scale (Page 12, lines 226-229) is commendable. However, the AER seems to be standardized only by age and sex. These rates would not be comparable or meaningful if they were not based on consistent population characteristics, particularly when considering potential differences in the risk factors for outcomes across the cohorts.

Thank you. The purpose of presenting AERs is to quantify the population-level impact of COVID-19, across strata defined by risk factors for COVID-19 and for the outcomes. As noted by reviewer, we standardised the AERs by age and sex. This is because the very different age-sex distributions of the three cohorts made unstandardized comparisons difficult to interpret and potentially misleading. In the methods section, we have clarified that the AERs were based on the maximally adjusted from each cohort. The amended text now reads "In each cohort absolute excess risks (AER) of any ATE and any VTE after COVID-19, weighted by the proportions of individuals in age and sex strata in the pre-vaccination cohort, were derived based on the maximally adjusted hazard ratios for the cohort" (page 21, lines 422-424). As shown in Figures S3 and S4, differences in estimated hazard ratios between subgroups defined by important risk factors were not substantial.

3. In Tables S3 and S6, the time period indicated as weeks "5-8" seems like it should be "5-28". Please verify.

Thank you for identifying this mistake. This has now been corrected in both table S3 and table S6.

4. The figures representing HRs across weeks could benefit from some enhancements. Currently, the information from different cohorts overlaps, making it challenging to be seen. Additionally, the estimates and confidence intervals for each time point are somewhat obscured due to the y-axis

range. Further refinement would aid in clarity and readability.

We made sure to make our figures as clear as possible. We think putting the results of the three cohorts in the same figure are essential to facilitate comparison. We are happy to discuss potential improvements to the figures with the editors.

REVIEWER COMMENTS

Reviewer #1 (Remarks to the Author):

My main concern about this study was that the "weeks 1-4" cohort included day 0 and it was unclear how quickly the hazard ratios fell. This was important because the abstract noted incidence was "substantially elevated during weeks 1-4", and there was no evidence that it really lasted that long. In this revision, the authors made two changes which address this problem. First, they separated day 0 and days 1-27 in the tables, figures, and text. Second, they added supplementary Table S4, which shows that the adjusted hazard ratios are indeed elevated throughout the four-week period.

The new Figure S6 is very helpful in understanding the timeline of the study.

Reviewer #2 (Remarks to the Author):

I agree with the authors' revisions and have scientific consistency suitable for publication.

Reviewer #3 (Remarks to the Author):

I would like to thank the authors for their detailed response to my comments. However, I noticed that the first two major comments have not been directly answered or addressed, which require further explanation.

1. I thank the authors for their further clarification regarding the sampling strategy. I understand the necessity of adjusting the analytic approach based on computational resources. However, the current writing suggests that the study assigns an IPW of 1 to those diagnosed with COVID-19, and the same IPW to individuals with an outcome event in the control group. Conversely, those without an event in the control group receive an IPW of $1/X$.

The response from the authors, along with the cited references, does not provide a valid explanation for assigning weights differently based on the outcome status in the control group. The principles of causal inference suggest that outcome information should not be used in constructing weighting.

Can the authors confirm if the weighted control group also has a similar event rate compared to the unweighted overall control group? If so, how would such sampling from random sample from the control? If not, the weighted control may not accurately represent the overall control, potentially leading to an underestimation of the risk difference between the COVID group and the control group.

While this might not impact the study's conclusion, as such bias is likely to skew results towards the null hypothesis, it may still be worth mentioning.

2. The rationale behind the authors' decision not to directly quantify the effect or association between vaccination and outcome, either as a hazard ratio or rate difference or even ratio of the ratio, remains unclear. Especially considering the study's focus on the impact of vaccination. It would be important to provide a direct comparison between the vaccinated group and the pre-vaccination or unvaccinated groups under similar baseline risk conditions. Without such comparison, one might argue that the observed differences in the hazard ratios in different COVID-19 groups compared to the control group are attributable to in the difference in group baseline characteristics modified the association between COVID-19 and the outcome.

Nature Communications manuscript NCOMMS-23-33238A
Authors' response to reviewers' comments

Author responses are in blue italic text.

We are grateful to the reviewers for their further comments. Reviewers 1 and 2 were happy with the revised manuscript, while Reviewer 3 had further comments and questions about the sampling-based analysis strategy and requested that we include direct comparisons between the three cohorts, which we have now done. We have provided a point-by-point response below.

REVIEWER COMMENTS

Reviewer #1 (Remarks to the Author):

My main concern about this study was that the “weeks 1-4” cohort included day 0 and it was unclear how quickly the hazard ratios fell. This was important because the abstract noted incidence was “substantially elevated during weeks 1-4”, and there was no evidence that it really lasted that long. In this revision, the authors made two changes which address this problem. First, they separated day 0 and days 1-27 in the tables, figures, and text. Second, they added supplementary Table S4, which shows that the adjusted hazard ratios are indeed elevated throughout the four-week period.

The new Figure S6 is very helpful in understanding the timeline of the study.

Thank you for your further review. We are pleased that we addressed your previous helpful comments.

Reviewer #2 (Remarks to the Author):

I agree with the authors' revisions and have scientific consistency suitable for publication.

Thank you - we are pleased that we addressed your previous helpful comments.

Reviewer #3 (Remarks to the Author):

I would like to thank the authors for their detailed response to my comments. However, I noticed that the first two major comments have not been directly answered or addressed, which require further explanation.

Thank you for your further comments. We provided a detailed response to your previous comments on the sampling-based analysis strategy, and we are sorry that you do not feel that your comments were directly answered or addressed. We were perhaps insufficiently clear that we do not agree with the following statements that were made in your first review:

“By including all non-COVID-19 participants who experienced the outcome and only randomly selecting non-COVID-19 participants without the outcome, there's a risk of introducing a significant bias. Such biases can't not be adequately address through weighting.”

We do not agree with this point, because the weighting exactly addresses the bias that would be introduced if we analysed the sampled individuals without weighting.

“This analytical approach artificially elevated the event rate within the non-COVID-19 (control) group, thereby create doubt on the accuracy of the results, especially when evaluating the effects of COVID-19 against this potentially skewed control group. It would been more appropriate to randomly select participants from the non-COVID-19 group without consideration of their outcome status.”

We do not agree with this point, because the weighting avoids artificially elevating the event rate within the non-COVID-19 (control) group. If 10% of the non-COVID-19 group who did not have the outcome were sampled then each of the sampled individuals receives a weight of 10 in the analysis. In the weighted analysis, the event rate within the sampled non-COVID-19 (control) group (including individuals with the outcome event) is identical to that in the whole non-COVID-19 group before sampling.

Randomly selecting participants from the non-COVID-19 group regardless of their outcome status would have been less efficient (leading to wider confidence intervals) because in time-to-event analyses the precision of estimated log hazard ratios depends primarily on the number of outcome events. Our approach maximizes precision by including all the outcome events.

1. I thank the authors for their further clarification regarding the sampling strategy. I understand the necessity of adjusting the analytic approach based on computational resources. However, the current writing suggests that the study assigns an IPW of 1 to those diagnosed with COVID-19, and the same IPW to individuals with an outcome event in the control group. Conversely, those without an event in the control group receive an IPW of $1/X$.

You are correct that those diagnosed with COVID-19 received an IPW of 1, regardless of whether they had an outcome event, because they were certain to be included in the analyses (they were sampled with probability 1). In addition, those not diagnosed with COVID-19 but who had an outcome event received an IPW of 1, because they were certain to be included in the analyses. All other individuals (those who were not diagnosed with COVID-19 and did not have an outcome event) were randomly sampled and received an IPW of $1/X$, where X was their probability of being sampled.

The response from the authors, along with the cited references, does not provide a valid explanation for assigning weights differently based on the outcome status in the control group.

The general approach to weighting used in our study was developed by Horvitz and Thompson more than 70 years ago (Horvitz DG, Thompson DJ. A generalization of sampling without replacement from a finite universe. Journal of the American Statistical Association 1952; 47: 663-685). This approach underpins much of survey statistics, because it provides a general means of analysing data from surveys in which subjects are sampled with unequal probabilities. Our study employs this standard approach from survey statistics.

The weights in our study are assigned identically for each sampled individual, as the inverse of the probability of being sampled. This leads to the weights described in the response paragraph immediately above. The explanation for assigning weights based on the outcome status in the control group is that in the weighted population the distribution of all variables included in the analysis is identical (other than through sampling variation) to that in the whole population. Weighting recreates the distribution of the exposure, outcome and confounding variables in the whole population.

The principles of causal inference suggest that outcome information should not be used in constructing weighting.

We do not agree that “The principles of causal inference suggest that outcome information should not be used in constructing weighting.” For example, this is exactly what is done in the well-established practice of using inverse-probability weights to account for informative censoring. As described in detail in pages 291-293 of What If by Hernán and Robins (https://www.hsph.harvard.edu/miguel-hernan/wp-content/uploads/sites/1268/2024/01/hernanrobins_WhatIf_2jan24.pdf), we construct a model for the probability of being censored over time, and inverse-probability weight follow-up in uncensored individuals so that it accounts for unobserved follow-up in censored individuals. Such procedures use outcome information, because censored individuals have not experienced the outcome. Individuals who experienced the outcome are unweighted in the analyses.

Can the authors confirm if the weighted control group also has a similar event rate compared to the unweighted overall control group? If so, how would such sampling from random sample from the control? If not, the weighted control may not accurately represent the overall control, potentially leading to an underestimation of the risk difference between the COVID group and the control group.

While this might not impact the study's conclusion, as such bias is likely to skew results towards the null hypothesis, it may still be worth mentioning.

We can confirm, and have checked, that the weighted control group has an identical event rate compared to the unweighted overall control group. We believe that the sampling and weighting used in our analyses does not lead to bias. We have conducted extensive simulation studies to confirm that estimates in weighted analyses are identical regardless of the sampling fractions used. A paper reporting these findings will be submitted soon.

We do not understand the question “If so, how would such sampling from random sample from the control?”. We think that this may have been mistyped.

2. The rationale behind the authors' decision not to directly quantify the effect or association between vaccination and outcome, either as a hazard ratio or rate difference or even ratio of the ratio, remains unclear. Especially considering the study's focus on the impact of vaccination. vaccinated group and the pre-vaccination or unvaccinated groups Without such comparison, one might argue that the observed differences in the hazard ratios in different COVID-19 groups compared to the control group are attributable to in the difference in group baseline characteristics modified the association between COVID-19 and the outcome.

We apologise that we did not fully address this point in our previous response. In our revised manuscript we now provide direct comparisons between the vaccinated group and the pre-vaccination or unvaccinated groups, as requested by the reviewer. The new Supplementary Table S4 now displays ratios (comparing the three groups) of the hazard ratios for associations between COVID-19 and arterial thrombotic events, and between COVID-19 and venous thrombotic events.

In the methods section (lines 424-426) we now write: “We calculated ratios of hazard ratios (with corresponding 95% CI) comparing the maximally adjusted HRs for ATE and VTE during weeks 1-4 and 5-28 between the three cohorts.”

In the results section (lines 181-186) we now write: “During weeks 1-4 the aHRs for ATE were substantially lower in the vaccinated cohort than in the unvaccinated or pre-vaccination cohorts (ratios of aHRs 0.28 (0.25-0.32) and 0.36 (0.33-0.38) respectively, Table S4). Although attenuated, aHRs remained lower in the vaccinated cohort than in the unvaccinated or pre-vaccination cohorts during weeks 5-28 (ratios of aHRs 0.70 (0.52-0.94) and 0.73 (0.66-0.82) respectively)”.

In lines 204-209 we now write: “During weeks 1-4 the aHRs for VTE were substantially lower in the vaccinated cohort than in the unvaccinated or pre-vaccination cohorts (ratios of aHRs 0.17 (0.15-0.19) and 0.24 (0.23-0.26) respectively, Table S4). Although attenuated, aHRs remained lower in the vaccinated cohort than in the unvaccinated or pre-vaccination cohorts during weeks 5-28 (ratios of aHRs 0.63 (0.49-0.80) and 0.61 (0.55-0.68) respectively).”

REVIEWERS' COMMENTS

Reviewer #3 (Remarks to the Author):

I thank the authors for providing measurements on the effects of vaccination and for clarifying that only those without an event in the control group were involved in estimating the probability. The authors have addressed all my comments, and I have no further comments.